# The Sparse Matrix-Based Random Projection: An Analysis of Matrix Sparsity for Classification

## Abstract

In the paper, we study the sparse $\{0, \pm 1\}$-matrix based random projection, which has been widely applied in classification tasks to reduce data dimensionality. For such sparse matrices, it is computationally interesting to explore the minimum number of nonzero entries $\pm 1$ required to achieve the best or nearly best classification performance. For this purpose, we analyze the impact of matrix sparsity on the $\ell_1$ distance between projected data points. Our analysis is grounded in the fundamental principle of Principle Component Analysis (PCA), which posits that larger variances between projected data points can better capture the variation inherent in the original data, thereby improving classification performance. Theoretically, the $\ell_1$ distance between projected data points is related not only to the sparsity of sparse matrices, but also to the distribution of original data. Without loss of generality, we consider two typical data distributions: the Gaussian mixture distribution and the two-point distribution, commonly employed in modeling real-world data distributions. Using these two data distributions, we estimate the $\ell_1$ distance between projected data points. It is found that the sparse matrices with only one or a few dozen nonzero entries per row, can provide comparable or even larger $\ell_1$ distances compared to denser matrices, provided the matrix size $m \geq \mathcal{O}(\sqrt{n})$. Accordingly, a similar performance trend should be observed in classification. This is confirmed with classification experiments on real data of different types, including image, text, gene and binary-quantized data.

## 1 Introduction

Random projection is a crucial unsupervised dimensionality reduction technique that projects high-dimensional data onto low-dimensional subspaces using random matrices (Johnson & Lindenstrauss, 1984). This projection method can approximately maintain the pairwise $\ell_2$ distance between original data points, preserving the structure of original data and making it suitable for classification tasks (Bingham & Mannila, 2001; Fradkin & Madigan, 2003; Wright et al., 2009). To ensure the preservation of $\ell_2$ distances, random projection matrices must adhere to specific distributions, such as Gaussian matrices (Dasgupta & Gupta, 1999) and sparse $\{0, \pm 1\}$-ternary matrices (hereinafter referred to as sparse matrices) (Achlioptas, 2003). In practical scenarios, sparse matrices are favored due to their significantly lower storage and computational complexity. Given that random projection is frequently used in computationally intensive, large-scale classification tasks, minimizing its complexity is highly desirable. To this end, we propose to explore the minimum number of nonzero entries $\pm 1$ required for the projected data to achieve the best or nearly best classification performance. To the best of our knowledge, this problem has not been previously investigated.

Existing research on random projection primarily focuses on exploring the distribution of random matrices that effectively maintain the distance preservation property. Specifically, this involves ensuring that the expectation of pairwise distances between original data points remains unchanged after random projection, while keeping the variance relatively small (Dasgupta & Gupta, 1999; Achlioptas, 2003). For sparse matrices with appropriately scaled entries, it has been demonstrated that the distance preservation property holds in the $\ell_2$ norm (Achlioptas, 2003; Li et al., 2006), but does *not* hold in the $\ell_1$ norm (Brinkman & Charikar, 2003; Li, 2007). It is important to note that while the $\ell_2$ distance preservation property allows random projection to be utilized in classification tasks, it does not lend itself well to analyzing the influence of matrix sparsity

(i.e., the number of nonzero entries) on subsequent classification accuracy. This is because classification accuracy is contingent on the discrimination between projected data points, rather than the preservation of the original data structure. For instance, it has been demonstrated that the $\ell_2$ distance preservation property deteriorates, as the matrix becomes sparser (Li et al., 2006), containing fewer nonzero entries $\pm 1$. However, empirically, it has been observed that a sparser matrix structure does not necessarily lead to poorer classification performance. In fact, very sparse matrices, such as those with only one or a few dozen nonzero entries per row, can achieve comparable or even superior classification results compared to denser matrices. In the paper, we demonstrate that this intriguing performance can be explained by examining the *variation* in the $\ell_1$ distance between projected data points, based on the principle that larger pairwise distances generally lead to better classification results. This principle is rooted in principal component analysis (PCA) (Jolliffe, 2002). Specifically, it is argued in PCA that projecting data onto larger principal components results in larger variances (achievable by larger pairwise distances) among the projected data points; and these larger variances are beneficial as they tend to better capture the variation (i.e., statistical information) inherent in the original data (Jolliffe & Cadima, 2016), thereby achieving better classification performance. This principle has been widely recognized and validated in various PCA applications, such as face recognition (Turk & Pentland, 1991).

In sparse matrix-based random projection, the $\ell_1$ distance between projected data points is influenced not only by the sparsity of random matrices, but also by the distribution of the original high-dimensional data. To examine the impact of matrix sparsity on the $\ell_1$ distance, it is necessary to initially model the distribution of the original data. For the sake of generality, we consider two common data distributions: the Gaussian mixture distribution and the two-point distribution. The Gaussian mixture distribution is frequently employed to represent the distribution of natural data (Torralba & Oliva, 2003; Weiss & Freeman, 2007) or their sparse feature transforms (Wainwright & Simoncelli, 1999; Lam & Goodman, 2000). On the other hand, the two-point distribution can model binary data distributions, which are often encountered in various quantization tasks (Gionis et al., 1999; Hubara et al., 2016; Yang et al., 2019). By leveraging these two fundamental distributions, our theoretical analysis results, as demonstrated later, can be extrapolated to a wide array of real-world data scenarios.

Given the two data distributions, by varying the sparsity of sparse matrices, we examine the *expected* $\ell_1$ distance between projected data points and derive two major results. First, sparse matrices with only one nonzero entry per row tend to achieve the maximum distance, as the discrimination between two classes of original data is sufficiently high. Second, as the matrix sparsity increases, the distance tends to converge to a constant value, with relatively minor convergence errors observed when sparse matrices contain only a few dozen nonzero entries per row. In summary, these findings suggest that sparse matrices with just one or a few dozen nonzero entries per row perform comparably, if not better, than denser matrices in terms of increasing the expected $\ell_1$ distance between projected data points. Consequently, these matrices are expected to exhibit similar performance trends in classification tasks. Note that our analysis is focused on the *expectation* of the $\ell_1$ distance. To practically approximate the expected distance using an actual matrix of size $m \times n$, we need $m \geq \mathcal{O}(\sqrt{n})$. In our experiments, the performance advantage of the sparse matrices mentioned above is validated by classification experiments on real data of different types, including the image, text, gene and binary-quantized data. The major contributions of the work can be summarized as follows:

- For sparse $\{0, \pm 1\}$-matrices based random projection, we for the first time investigate the impact of matrix sparsity on classification performance. Instead of focusing on traditional distance preservation, our analysis examines random projection from the perspective of *distance variation*. This analysis is grounded in the PCA principle (Jolliffe & Cadima, 2016; Turk & Pentland, 1991), which suggests that a larger distance between projected data points can more accurately reflect the variation in the original data, thereby yielding superior classification results.

- Through theoretical analysis, we have demonstrated that sparse matrices containing just one or a few dozen nonzero entries per row tend to exhibit comparable or even superior classification performance compared to denser matrices. This is achieved when the original data can be effectively modeled using the Gaussian mixture or two-point distribution, and the matrices adhere to the size constraint of $m \geq \mathcal{O}(\sqrt{n})$. This finding suggests that by employing such sparse matrices, we can notably

reduce the complexity of random projection matrices without compromising, and sometimes even improving classification accuracy.

- The theoretical results are perfectly validated through numerical simulations and practical experiments. The remarkable *consistency* observed between theory and empirical results can be ascribed to the robust generalizability of the two aforementioned distributions employed to characterize the original data. These distributions have been recognized for their effectiveness in modeling diverse data types (Torralba & Oliva, 2003; Weiss & Freeman, 2007; Wainwright & Simoncelli, 1999; Lam & Goodman, 2000), thus underpinning the practical applicability of our theoretical findings.

## 2 Problem Formulation

Consider the random projection of two data points $\boldsymbol{h}, \boldsymbol{h}' \in \mathbb{R}^n$ over a sparse random matrix $\boldsymbol{R} \in \{0, \pm 1\}^{m \times n}$. For the matrix $\boldsymbol{R}$, we attempt to estimate the minimum number of nonzero entries $\pm 1$ that enables maximizing the expected $\ell_1$ distance $\mathbb{E}\|\boldsymbol{R}\boldsymbol{x}\|_1$ between the projections of $\boldsymbol{h}$ and $\boldsymbol{h}'$, where $\boldsymbol{x} = \boldsymbol{h} - \boldsymbol{h}'$. As discussed before, the maximum $\mathbb{E}\|\boldsymbol{R}\boldsymbol{x}\|_1$ is expected to provide the best classification performance. To determine the minimum sparsity, we need to investigate the changing trend of $\mathbb{E}\|\boldsymbol{R}\boldsymbol{x}\|_1$ against the varying sparsity of $\boldsymbol{R}$. It can seen that the estimation depends on the distributions of the matrix $\boldsymbol{R}$ and the data $\boldsymbol{h}$. So in the following, we first model the distributions of sparse matrices and real data, and then give the $\ell_1$ estimation model.

**Notation.** Throughout the work, we typically denote a matrix by a bold upper-case letter $\boldsymbol{R} \in \mathbb{R}^{m \times n}$, a vector by a bold lower-case letter $\boldsymbol{r} = (r_1, r_2, ..., r_n)^\top \in \mathbb{R}^n$, and a scalar by a lower-case letter $r_i$ or $r$. Sometimes, we use the bold letter $\boldsymbol{r}_i \in \mathbb{R}^n$ to denote the $i$-th row of $\boldsymbol{R} \in \mathbb{R}^{m \times n}$. For ease of presentation, we defer all proofs to Appendix A.

### 2.1 The distribution of sparse matrices

The sparse random matrix $\boldsymbol{R}$ we aim to study is specified in Definition 1, which has the parameter $k$ counting the number of nonzero entries per row, and is simply called $k$-sparse to distinguish between the matrices of different sparsity. Instead of the form $\boldsymbol{R} \in \{0, \pm 1\}^{m \times n}$, in the definition we introduce a scaling parameter $\sqrt{\frac{n}{mk}}$ to make the matrix entries have zero mean and unit variance. With this distribution, the matrix will hold the $\ell_2$ distance preservation property, that is, keeping the expected $\ell_2$ distance between original data points unchanged after random projection (Achlioptas, 2003). Note that the scaling parameter can be omitted in practical applications for easier computation; and the omitting will not change the relative distances between projected data points, thus not affecting the follow-up classification performance.

**Definition 1** ($k$-sparse random matrix)**.** A $k$-sparse random matrix $\boldsymbol{R} \in \{0, \pm\sqrt{\frac{n}{mk}}\}^{m \times n}$ is defined to be of the following structure properties:

- its each row vector $\boldsymbol{r} \in \{0, \pm\sqrt{\frac{n}{mk}}\}^n$ contains exactly $k$ nonzero entries, $1 \leq k \leq n$;

- the positions of $k$ nonzero entries are arranged uniformly at random;

- each nonzero entry takes the bipolar values $\pm\sqrt{\frac{n}{mk}}$ with equal probability.

### 2.2 The distribution of original data

For the original high-dimensional data $\boldsymbol{h} \in \mathbb{R}^n$, as discussed before, we investigate two typical distributions, the two-point distribution and the Gaussian mixture distribution. Considering the expected $\ell_1$ distance $\mathbb{E}\|\boldsymbol{R}\boldsymbol{x}\|_1$ is directly related to the pairwise difference $\boldsymbol{x}$ between two original data $\boldsymbol{h}$ and $\boldsymbol{h}'$, namely $\boldsymbol{x} = \boldsymbol{h} - \boldsymbol{h}' = (x_1, x_2, \ldots, x_n)^\top$, we describe the distribution of $\boldsymbol{x}$ for the original data $\boldsymbol{h}$ with the given two distributions.

### 2.2.1 Two-point distribution

Suppose that the two high-dimensional data vectors $\boldsymbol{h}, \boldsymbol{h}' \in \{\mu_1, \mu_2\}^n$ have their each entry independently following a two-point distribution, where $\mu_1$ and $\mu_2$ are two arbitrary constants. Then the difference $\boldsymbol{x}$ between $\boldsymbol{h}$ and $\boldsymbol{h}'$ has its each entry $x_i$ independently following a ternary discrete distribution

$$x_i \sim \mathcal{T}(\mu, p, q) \tag{1}$$

with the probability mass function $t \in \{-\mu, 0, \mu\}$ under the probabilities $\{q, p, q\}$, where $\mu = \mu_1 - \mu_2$ and $p + 2q = 1$.

### 2.2.2 Gaussian mixture distribution

For the two data vectors $\boldsymbol{h}, \boldsymbol{h}' \in \mathbb{R}^n$, each corresponding pair of elements $h_i$ and $h_i'$ will be drawn either from the same distribution, or from different ones. To reflect this fact, we employ a two-component Gaussian mixture model for the elements: $h_i, h_i' \sim \Sigma_{i=1}^2 \phi_i N(\mu_i, \sigma^2)$, where $\mu_1 \neq \mu_2$, $\phi_1 + \phi_2 = 1$, and $\phi_i \geq 0$. When both $h_i$ and $h_i'$ are from the same distribution, either $\mathcal{N}(\mu_1, \sigma^2)$ or $\mathcal{N}(\mu_2, \sigma^2)$, their difference $x_i = h_i - h_i'$ follows $\mathcal{N}(0, 2\sigma^2)$; otherwise, $x_i$ follows either $\mathcal{N}(\mu_1 - \mu_2, 2\sigma^2)$ or $\mathcal{N}(-\mu_1 + \mu_2, 2\sigma^2)$. Consequently, the differences $x_i$ form a three-component Gaussian mixture, which can be formally modeled as

$$x_i \sim \mathcal{M}(\mu, \sigma^2, p, q) \tag{2}$$

with the probability density function

$$f(t) = p f_{\mathcal{N}}(t; 0, \sigma^2) + q f_{\mathcal{N}}(t; \mu, \sigma^2) + q f_{\mathcal{N}}(t; -\mu, \sigma^2) \tag{3}$$

where $f_{\mathcal{N}}(t; \mu, \sigma^2)$ denotes the density function of $t \sim \mathcal{N}(\mu, \sigma^2)$, and $p$ and $q$ represent the mixture weights of three components, with $p + 2q = 1$, and $p, q \geq 0$. Note that a smaller $p$ (equivalently, a larger $q$) indicates a greater number of nonzero entries in $x_i$, suggesting a higher degree of discrimination between $h_i$ and $h_i'$.

## 2.3 The $\ell_1$ distance estimation model

With the distributions defined for the original data points $\boldsymbol{h}, \boldsymbol{h}' \in \mathbb{R}^n$ and the $k$-sparse random matrix $\boldsymbol{R} \in \{0, \pm\sqrt{\frac{n}{mk}}\}^{m \times n}$, our goal is to analyze the changing of the expected $\ell_1$ distance $\mathbb{E}\|\boldsymbol{R}\boldsymbol{x}\|_1$ (with $\boldsymbol{x} = \boldsymbol{h} - \boldsymbol{h}'$) against varying matrix sparsity $k$, and determine the minimum sparsity $k$ that corresponds to the maximum or nearly maximum $\mathbb{E}\|\boldsymbol{R}\boldsymbol{x}\|_1$. Notice that we have $\mathbb{E}\|\boldsymbol{R}\boldsymbol{x}\|_1 = m\mathbb{E}|\boldsymbol{r}^\top \boldsymbol{x}|$, since each row $\boldsymbol{r} \in \mathbb{R}^n$ of $\boldsymbol{R}$ follows an independent and identical distribution by Definition 1. This equivalence suggests that $\mathbb{E}|\boldsymbol{r}^\top \boldsymbol{x}|$ will share the same changing trend with $\mathbb{E}\|\boldsymbol{R}\boldsymbol{x}\|_1$, when varying $k$. Then for ease of analysis, instead of $\mathbb{E}\|\boldsymbol{R}\boldsymbol{x}\|_1$, in the following we choose to investigate the changing of $\mathbb{E}|\boldsymbol{r}^\top \boldsymbol{x}|$ against varying $k$.

# 3 The $\ell_1$ Distance Estimation with Two-Point Distributed Data

In this section, we investigate the changing of $\mathbb{E}|\boldsymbol{r}^\top \boldsymbol{x}|$ against varying matrix sparsity $k$, provided that the original data $\boldsymbol{h}, \boldsymbol{h}'$ are drawn from two-point distributions, such that their difference $\boldsymbol{x} = \boldsymbol{h} - \boldsymbol{h}'$ has i.i.d. entries $x_i \sim \mathcal{T}(\mu, p, q)$, as specified in (1).

## 3.1 Theoretical analysis

**Theorem 1.** Let $\boldsymbol{r}$ be a row vector of a $k$-sparse random matrix $\boldsymbol{R} \in \{0, \pm\sqrt{\frac{n}{mk}}\}^{m \times n}$, and $\boldsymbol{x} \in \mathbb{R}^n$ with i.i.d. entries $x_i \sim \mathcal{T}(\mu, p, q)$. It can be derived that

$$\mathbb{E}|\boldsymbol{r}^\top \boldsymbol{x}| = 2\mu \sqrt{\frac{n}{mk}} \sum_{i=0}^{k} C_k^i p^i q^{k-i} \left\lceil \frac{k-i}{2} \right\rceil C_{k-i}^{\lceil \frac{k-i}{2} \rceil} \tag{4}$$

and

$$\mathrm{Var}(|\boldsymbol{r}^\top \boldsymbol{x}|) = \frac{2q\mu^2 n}{m} - \frac{4\mu^2 n}{mk} \left( \sum_{i=0}^{k} C_k^i p^i q^{k-i} \left\lceil \frac{k-i}{2} \right\rceil C_{k-i}^{\lceil \frac{k-i}{2} \rceil} \right)^2 \tag{5}$$

where $C_k^i$ is a binomial coefficient $\binom{k}{i}$ and $\lceil \alpha \rceil = \min\{\beta : \beta \geq \alpha, \beta \in \mathbb{Z}\}$. By (4), $\mathbb{E}|\boldsymbol{r}^\top \boldsymbol{x}|$ satisfies the following two properties:

(P1) When $p \leq 0.188$, $\mathbb{E}|\boldsymbol{r}^\top \boldsymbol{x}|$ has its maximum at $k = 1$.

(P2) When $k \to \infty$, $\mathbb{E}|\boldsymbol{r}^\top \boldsymbol{x}|$ converges to a constant:

$$\lim_{k \to \infty} \frac{\sqrt{m}}{\mu\sqrt{n}} \mathbb{E}|\boldsymbol{r}^\top \boldsymbol{x}| = 2\sqrt{q/\pi}, \tag{6}$$

which has the convergence error for finite $k$ upper-bounded by

$$\left| \frac{\sqrt{m}}{\mu\sqrt{n}} \mathbb{E}|\boldsymbol{r}^\top \boldsymbol{x}| - 2\sqrt{q/\pi} \right| \leq \frac{\sqrt{\pi} + \sqrt{2}}{\sqrt{\pi k}}. \tag{7}$$

**Remarks of Theorem 1:** In P1 and P2, we characterize the changing trends of $\mathbb{E}|\boldsymbol{r}^\top \boldsymbol{x}|$ against varying matrix sparsity $k$, which are further discussed as follows.

- By P1, $\mathbb{E}|\boldsymbol{r}^\top \boldsymbol{x}|$ can achieve its maximum value at $k = 1$, if the probability $p$ of $x_i = 0$ is sufficiently small, namely $p \leq 0.188$. As discussed in Section 2.2.2, this condition indicates that the difference $\boldsymbol{x}$ between two data points $\boldsymbol{h}$ and $\boldsymbol{h}'$ should contain a sufficient number of nonzero entries, suggesting that the two data points $\boldsymbol{h}$ and $\boldsymbol{h}'$ should be sufficiently distinct from each other. Then we can say that for two data distributions that exhibit sufficiently high discrimination, the best classification performance can be attained using very sparse random matrices with sparsity $k = 1$. This is due to the maximum $\mathbb{E}|\boldsymbol{r}^\top \boldsymbol{x}|$ being achieved at $k = 1$.

- By P2, $\mathbb{E}|\boldsymbol{r}^\top \boldsymbol{x}|$ will converge to a constant that depends on the data distribution and matrix size, as $k$ tends to infinity. Note that in (6) we describe the convergence with $\mathbb{E}|\boldsymbol{r}^\top \boldsymbol{x}|/(\mu\sqrt{n/m})$ instead of $\mathbb{E}|\boldsymbol{r}^\top \boldsymbol{x}|$, in terms of the fact that both formulas share the same changing trend against varying $k$, but the former has fewer parameters, only involving $k$ and $p$. Moreover, it is noteworthy that the convergence error, namely the difference between the values of $\mathbb{E}|\boldsymbol{r}^\top \boldsymbol{x}|$ with finite $k$ and infinite $k$, is upper-bounded in (7), and the bound indicates a convergence speed $\mathcal{O}(1/\sqrt{k})$. By the bound (7), it is easy to further derive that

$$\frac{\left| \frac{\sqrt{m}}{\mu\sqrt{n}} \mathbb{E}|\boldsymbol{r}^\top \boldsymbol{x}| - 2\sqrt{q/\pi} \right|}{2\sqrt{q/\pi}} \leq \eta, \quad \text{if } k \geq \frac{(\sqrt{\pi} + \sqrt{2})^2}{4q\eta^2} \tag{8}$$

where $\eta$ can be an arbitrary positive constant. It is seen that $\eta$ sets an upper bound for the ratio between the convergence error with the convergence value (hereinafter referred to as the convergence ratio error). For any arbitrarily small upper bound $\eta$, as shown in (8), there exists a corresponding minimum sparsity $k$ required to maintain this bound. When $k$ falls within this specified range, $\mathbb{E}|r^\top x|$ assumes similar values, suggesting that these $k$ values should result in similar classification performance. Consequently, sparse matrices with small $k$ values (around the minimum threshold) exhibit classification performance that is that is on par with denser matrices possessing larger $k$ values. Note that the lower bound of $k$ derived in (8) contains slack, and in practice it tends to be significantly smaller, as illustrated in the subsequent numerical analysis.

## 3.2 Numerical analysis

To more closely examine the changing trend of $\mathbb{E}|\boldsymbol{r}^\top \boldsymbol{x}|/(\mu\sqrt{n/m})$ against varying matrix sparsity $k$ (derived in P1 and P2), we directly compute the value of $\mathbb{E}|\boldsymbol{r}^\top \boldsymbol{x}|/(\mu\sqrt{n/m})$ by (4). Note that besides the parameter $k$, $\mathbb{E}|\boldsymbol{r}^\top \boldsymbol{x}|/(\mu\sqrt{n/m})$ also involves the parameter $p$, the probability of $x_i = 0$ as specified in (1). So we investigate $\mathbb{E}|\boldsymbol{r}^\top \boldsymbol{x}|/(\mu\sqrt{n/m})$ over $k \in [1, 500]$ for different $p \in (0, 1)$. For brevity, we here only provide

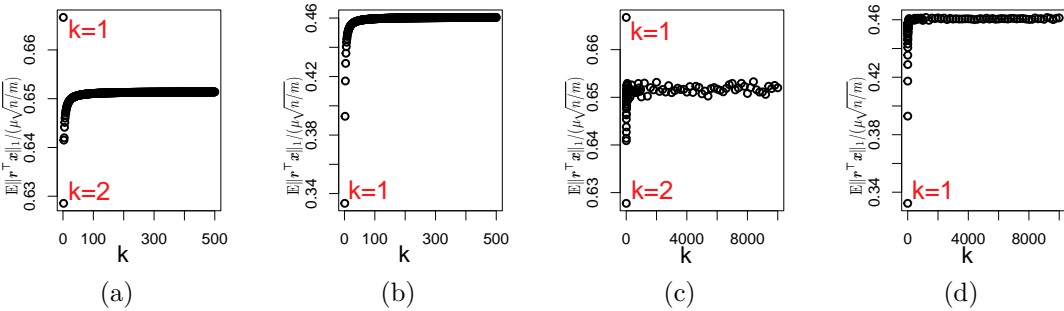

Figure 1: The value of $\mathbb{E}|\boldsymbol{r}^\top\boldsymbol{x}|/(\mu\sqrt{n/m})$ calculated by (4) with $p=1/3$ (a) and $p=2/3$ (b), and estimated by statistical simulation with $p=1/3$ (c) and $p=2/3$ (d), provided $x_i \sim \mathcal{T}(\mu, p, q)$, $\mu = 1$.

the results of $p = 1/3$ and $2/3$ in Figs. 1 (a) and (b), see the supplement for more results. By the numerical analysis results, we revisit the two changing trends described in P1 and P2 and obtain more positive conclusions:

(P3) When $p \leq 1/3$, such as the case of $p = 1/3$ shown in Fig. 1(a), $\mathbb{E}|\boldsymbol{r}^\top\boldsymbol{x}|/(\mu\sqrt{n/m})$ tends to achieve its maximum value at $k = 1$, but at other larger $k$ when $p > 1/3$, such as the case of $p = 2/3$ illustrated in Fig. 1(b). The results indicate that to maximize $\mathbb{E}|\boldsymbol{r}^\top\boldsymbol{x}|/(\mu\sqrt{n/m})$ at $k = 1$, the condition $p \in [0, 1/3)$ is sufficient, which is broader than the theoretical requirement $p \in [0, 0.188)$ derived from P1. Recall that a wider range of $p$ allows for a larger space of data as modeled in (3). This suggests that the desired property of maximizing $\mathbb{E}|\boldsymbol{r}^\top\boldsymbol{x}|/(\mu\sqrt{n/m})$ at $k = 1$ can be achieved over a wider range of $p$ values than what was theoretically predicted. To achieve a small $p$ within the range of $p \in [0, 1/3)$, as pointed out in Section 2.2.2, the original data points $\boldsymbol{h}$ and $\boldsymbol{h}'$ need to exhibit sufficiently high discrimination between them.

(P4) With the increasing of $k$, as the two cases of $p = 1/3$ and $2/3$ shown in Figs. 1(a) and (b), $\mathbb{E}|\boldsymbol{r}^\top\boldsymbol{x}|/(\mu\sqrt{n/m})$ tends to converge to the limit value $2\sqrt{q/\pi}$ derived in (6), where $q = (1 - p)/2$. Furthermore, it can be seen that small convergence errors will be achieved, when $k$ is very small, typically in the range of a few tens. For instance, in Fig. 2(a) we derive the convergence error ratios as defined in (8), which give the values close to zero when $k \geq 20$ and $p$ is relatively small. Recall that the small $p$ value implies that the original data have high discrimination between each other. With the decreasing of data discrimination, we should need larger $k$ to achieve small convergence errors.

In the analysis of the expected distance $\mathbb{E}|\boldsymbol{r}^\top\boldsymbol{x}|$, the influence of the variance $\mathrm{Var}(|\boldsymbol{r}^\top\boldsymbol{x}|)$ in (5) should be considered. Statistically, a lower variance $\mathrm{Var}(|\boldsymbol{r}^\top\boldsymbol{x}|)$ indicates a higher probability that the actual distance $|\boldsymbol{r}^\top\boldsymbol{x}|$ of a single matrix closely approximates its expected value $\mathbb{E}|\boldsymbol{r}^\top\boldsymbol{x}|$. Also, this implies a higher consistence between theoretical and practical results. By computing (5), we observe a trend similar to $\mathbb{E}|\boldsymbol{r}^\top\boldsymbol{x}|$: as $k$ increases, $\mathrm{Var}(|\boldsymbol{r}^\top\boldsymbol{x}|)$ tends to quickly converge to a constant value (see the supplement for further details). This suggests that $\mathrm{Var}(|\boldsymbol{r}^\top\boldsymbol{x}|)$ varies minimally across different $k$ values. Therefore, the probability of $|\boldsymbol{r}^\top\boldsymbol{x}|$ approximating $\mathbb{E}|\boldsymbol{r}^\top\boldsymbol{x}|$ remains consistent for various $k$, enabling us to use $\mathbb{E}|\boldsymbol{r}^\top\boldsymbol{x}|$ to reasonably estimate and compare the distances $|\boldsymbol{r}^\top\boldsymbol{x}|$ of actual matrices across different $k$.

### 3.3 Statistical simulation

To verify the correctness of Theorem 1, including the expression (4) of $\mathbb{E}|\boldsymbol{r}^\top\boldsymbol{x}|$ and its two properties P1 and P2, we here estimate the expectation value $\mathbb{E}|\boldsymbol{r}^\top\boldsymbol{x}|/(\mu\sqrt{n/m})$ (against varying $k$) by averaging over the statistically generated samples of $\boldsymbol{r}$ and $\boldsymbol{x}$. If the theorem results are correct, the statistical simulation results should be consistent with the numerical analysis results P3 and P4 (derived by Theorem 1). The simulation is introduced as follows. First, we randomly generate $10^6$ pairs of $\boldsymbol{r}$ and $\boldsymbol{x}$ from their respective distributions, i.e. $\boldsymbol{r} \in \{0, \pm\sqrt{\frac{n}{mk}}\}^n$ with $k$ nonzero entries randomly distributed, and $\boldsymbol{x}$ with i.i.d. $x_i \sim \mathcal{T}(\mu, p, q)$. Then,

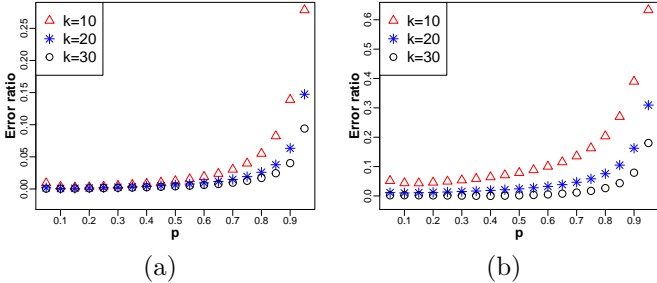

Figure 2: The convergence error ratios of three different $k \in \{10, 20, 30\}$ over varying $p$ are derived for two-point distributed data (a) and Gaussian mixture data (b), by computing the left side of the inequality of $\eta$ shown respectively in Eqs. (8) and (14).

the average value of $|r^\top x|/(\mu\sqrt{n/m})$ is derived as the final estimate of $\mathbb{E}|r^\top x|/(\mu\sqrt{n/m})$. The parameters for the distributions of $r$ and $x$ are set as follows: $m = 1$, $n = 10^4$, $\mu = 1$, and $p = 1/3$ or $2/3$. The data dimension $n = 10^4$ allows us to increase $k$ from 1 to $10^4$. The average value of $|r^\top x|/(\mu\sqrt{n/m})$ at each $k$ is provided in Figs. 1(c) and (d), respectively for the cases of $p = 1/3$ and $2/3$. Note that the choices of $m$, $n$ and $\mu$ will not affect the changing trend of $\mathbb{E}|r^\top x|/(\mu\sqrt{n/m})$ against $k$. Comparing the numerical analysis results with the simulation results presented in Fig. 1, specifically contrasting (a) vs. (c) and (b) vs. (d), it is seen that both sets of results exhibit similar trends in the variation of $\mathbb{E}|r^\top x|/(\mu\sqrt{n/m})$. The similarity between them validates Theorem 1, as well as the numerical analysis results P3 and P4.

Moreover, it is noteworthy that what we estimate is an *expected* distance $\mathbb{E}\|Rx\|_1$ (equivalently, $m\mathbb{E}|r^\top x|$), rather than the actual distance $\|Rx\|_1$ we will derive with a given matrix sample. To approximate the expected distance, by Property 1 the actual matrices should have the size of $m \geq \mathcal{O}(\sqrt{n})$.

**Property 1.** Let $r_i$ be the $i$-th row of a $k$-sparse random matrix $R \in \{0, \pm\sqrt{\frac{n}{mk}}\}^{m \times n}$, and $x \in \mathbb{R}^n$ with i.i.d. entries $x_i \sim \mathcal{T}(\mu, p, q)$. Suppose $z = \frac{1}{m}\|Rx\|_1 = \frac{1}{m}\sum_{i=1}^m |r_i^\top x|$. For arbitrary small $\varepsilon, \delta > 0$, we have the probability $\Pr\{|z - \mathbb{E}z| \leq \varepsilon\} \geq 1 - \delta$, if $\frac{m^2}{m+1} \geq \frac{q\mu^2 n}{\varepsilon^2\delta}$; and the condition can be relaxed to $m^2 \geq \frac{2q\mu^2 n}{\varepsilon^2\delta}$, for a given $x$.

## 4 The $\ell_1$ Distance Estimation with Gaussian Mixture Data

In this section, we consider the case that the original data $h$, $h'$ are drawn from Gaussian mixture distributions, such that their difference $x = h - h'$ has i.i.d. entries $x_i \sim \mathcal{M}(\mu, \sigma^2, p, q)$, as specified in (2). With such data, the changing of $\mathbb{E}|r^\top x|$ against varying matrix sparsity $k$ is analyzed.

### 4.1 Theoretical analysis

**Theorem 2.** Let $r$ be a row vector of a $k$-sparse random matrix $R \in \{0, \pm\sqrt{\frac{n}{mk}}\}^{m \times n}$, and $x \in \mathbb{R}^n$ with i.i.d. entries $x_i \sim \mathcal{M}(\mu, \sigma^2, p, q)$. It can be derived that

$$\mathbb{E}|r^\top x| = 2\mu\sqrt{\frac{n}{mk}}T_1 + \sigma\sqrt{\frac{2n}{\pi m}}T_2 - 2\mu\sqrt{\frac{n}{mk}}T_3 \tag{9}$$

$$T_1 = \sum_{i=0}^{k} C_k^i p^i q^{k-i} \left\lceil \frac{k-i}{2} \right\rceil C_{k-i}^{\lceil \frac{k-i}{2} \rceil}$$

$$T_2 = \sum_{i=0}^{k} C_k^i p^i q^{k-i} \sum_{j=0}^{k-i} C_{k-i}^j e^{-\frac{(k-i-2j)^2\mu^2}{2k\sigma^2}}$$

$$T_3 = \sum_{i=0}^{k} C_k^i p^i q^{k-i} \sum_{j=0}^{k-i} C_{k-i}^j \Phi\left(-\frac{|k-i-2j|\mu}{\sqrt{k}\sigma}\right)$$

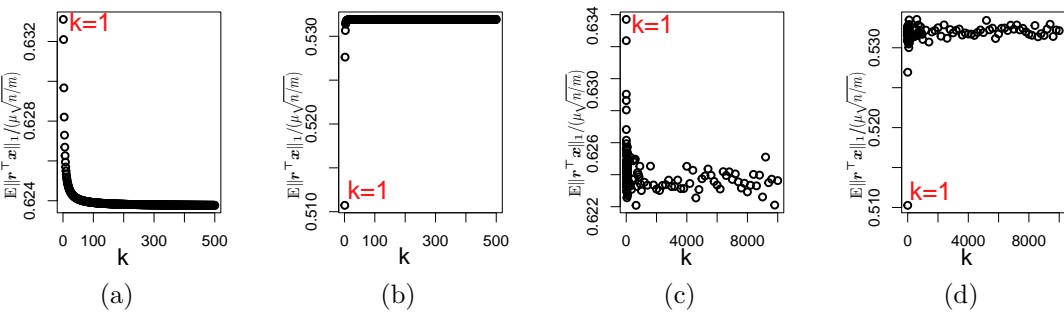

Figure 3: The value of $\mathbb{E}|\boldsymbol{r}^\top \boldsymbol{x}|/\sqrt{n/m}$ calculated by (9) with $p = 1/2$ (a) and $p = 2/3$ (b), and estimated by statistical simulation with $p = 1/2$ (c) and $p = 2/3$ (d), provided $x_i \sim \mathcal{M}(p, q, \mu, \sigma^2)$, $\mu = 1$ and $\sigma = 1/3$.

and

$$\mathrm{Var}(|\boldsymbol{r}^\top \boldsymbol{x}|) = \frac{n}{m}(\sigma^2 + 2q\mu^2) - \left(\mathbb{E}|\boldsymbol{r}^\top \boldsymbol{x}|_1\right)^2 \tag{10}$$

where $\Phi(\cdot)$ is the distribution function of $\mathcal{N}(0,1)$. Further, we have

$$\mathbb{E}|\boldsymbol{r}^\top \boldsymbol{x}| \leq \mu\sqrt{\frac{n}{m}} + \sigma\sqrt{\frac{2n}{\pi m}}, \tag{11}$$

and

$$\lim_{k \to \infty} \frac{\sqrt{m}}{\mu\sqrt{n}}\mathbb{E}|\boldsymbol{r}^\top \boldsymbol{x}| = \sqrt{\frac{2}{\pi}(\sigma^2 + 2q\mu^2)} \tag{12}$$

which has the convergence error for finite $k$ upper-bounded by

$$\left|\frac{\sqrt{m}}{\mu\sqrt{n}}\mathbb{E}|\boldsymbol{r}^\top \boldsymbol{x}| - \sqrt{2(\sigma^2 + 2q\mu^2)/\pi}\right| \leq \frac{4\sigma^3\left[p + 2q(1 + \mu^2/\sigma^2)^{3/2}\right]}{(\sigma^2 + 2q\mu^2)\sqrt{\pi k}} + \frac{\sqrt{2}[3\sigma^4 + 2q(6\sigma^2\mu^2 + \mu^4)]}{\sqrt{(\sigma^2 + 2q\mu^2)\pi k}}. \tag{13}$$

**Remarks of Theorem 2:** In Eqs. (12) and (13), we obtain two results similarly as in P2 of Theorem 1. First, $\mathbb{E}|\boldsymbol{r}^\top \boldsymbol{x}|$ converges to a constant with speed $\mathcal{O}(1/\sqrt{k})$. Second, by (13), we can derive the lower bound of $k$ that ensures the convergence error ratio upper-bounded by any given constant $\eta$:

$$\frac{\left|\frac{\sqrt{m}}{\mu\sqrt{n}}\mathbb{E}|\boldsymbol{r}^\top \boldsymbol{x}| - \sqrt{2(\sigma^2 + 2q\mu^2)/\pi}\right|}{\sqrt{2(\sigma^2 + 2q\mu^2)/\pi}} \leq \eta, \quad \text{if} \quad k \geq \left(\frac{4\sigma^3[p + 2q(1 + \mu^2/\sigma^2)^{3/2}]}{(\sigma^2 + 2qu^2)^{3/2}\sqrt{2}\eta} + \frac{3\sigma^4 + 2q(6\sigma^2\mu^2 + \mu^4)}{(\sigma^2 + 2q\mu^2)\eta}\right)^2. \tag{14}$$

As discussed in the remarks of Theorem 1, the lower bound of $k$ derived in Eq. (14) for a given small $\eta$ indicates a small matrix sparsity $k$ that performs comparably to the other larger sparsity $k$ in classification. Usually, as shown in the subsequent numerical analysis, the lower bound of $k$ is small, implying very sparse matrix structures. Moreover, the numerical analysis demonstrates that similarly to P1 of Theorem 1, $\mathbb{E}|\boldsymbol{r}^\top \boldsymbol{x}|$ in (12) also has its maximum attained at $k = 1$, when the data distribution parameter $p$ specified in (2) takes relatively small values.

## 4.2 Numerical analysis

In this part, we directly compute the value of $\mathbb{E}|\boldsymbol{r}^\top \boldsymbol{x}|/(\mu\sqrt{n/m})$ by (9). Note that $\mathbb{E}|\boldsymbol{r}^\top \boldsymbol{x}|/(\mu\sqrt{n/m})$ involves four parameters: $k$, $p$, $\mu$, and $\sigma$. In computing (9), we fix $\mu = 1$ and vary other parameters in the ranges of $\sigma/\mu \in (0, 1/3)$, $p \in (0, 1)$ and $k \in [1, 500]$. Here, we upper bound the value range of $\sigma/\mu$ by $1/3$ for easy simulation. Empirically, the changing trend of $\mathbb{E}|\boldsymbol{r}^\top \boldsymbol{x}|/(\mu\sqrt{n/m})$ is not sensitive to the varying of $\sigma/\mu$, but sensitive to $p$, i.e. the probability of each entry $x_i$ of the data difference $\boldsymbol{x}$ taking zero value, as specified in (2). In Figs. 3(a) and (b), we provide two typical results of $p = 1/2$ and $2/3$, and observe two properties similar to the previous P3 and P4:

(P5) When $p \leq 1/2$, such as the case of $p = 1/2$ and $\sigma/\mu = 1/3$ shown in Fig. 3(a), $\mathbb{E}|\boldsymbol{r}^\top \boldsymbol{x}|/\mu\sqrt{n/m}$ tends to obtain its maximum at $k = 1$, but at other larger $k$ when $p > 1/2$, such as the case of $p = 2/3$ and $\sigma/\mu = 1/3$ shown in Fig. 3(b). It can be seen that the upper bound of $p$ obtained here for Gaussian mixture data is relaxed from $2/3$ to $1/2$ compared to the bound derived in P3 for two-point distributed data. This implies a wider range of data distributions that enable obtaining the maximum $\mathbb{E}|\boldsymbol{r}^\top \boldsymbol{x}|/\mu\sqrt{n/m}$ at $k = 1$.

(P6) With the increasing of $k$, as the two results shown in Fig. 3(a) and (b), $\mathbb{E}|\boldsymbol{r}^\top \boldsymbol{x}|/(\mu\sqrt{n/m})$ converges to the limit value derived in (12). Similarly to the convergence discussed in P4 for two-distributed data, the convergence error ratio defined in (14) can approach zero with small $k$, such as $k = 20$ shown in Fig. 2(b), especially when $p$ is relatively small, namely the original data having relatively high discrimination.

For P5 and P6, their similarity to P3 and P4 is not surprising, since the two-point distribution $x_i \sim \mathcal{T}(\mu, p, q)$ can be viewed as an extreme case of the Gaussian mixture distribution $x_i \sim \mathcal{M}(\mu, \sigma^2, p, q)$ with $\sigma \to 0$. Thanks to the good generalizability of Gaussian mixture models, as will be seen in our experiments, the two properties analyzed above apply to a variety of real-world data.

Again note that we should have the matrix row size $m \geq \mathcal{O}(\sqrt{n})$, such that the actual distance $\|\boldsymbol{R}^\top \boldsymbol{x}\|_1$ computed with a single random matrix sample can approximate the expected distance $\mathbb{E}\|\boldsymbol{R}^\top \boldsymbol{x}\|_1$ (equivalently $m\mathbb{E}|\boldsymbol{r}^\top \boldsymbol{x}|$) derived with (9). The analysis is similar to Property 1, thus omitted here.

### 4.3 Statistical simulation

Similarly as in Section 3.3, we here verify the correctness of Theorem 2, including the expression (9) of $\mathbb{E}|\boldsymbol{r}^\top \boldsymbol{x}|$ and its convergence (12) by performing statistical simulation on $\boldsymbol{x}$ and $\boldsymbol{r}$. The simulation results should agree with the numerical analysis results P5 and P6, if the theorem is correct. In the simulation, we estimate the value of $\mathbb{E}|\boldsymbol{r}^\top \boldsymbol{x}|/\sqrt{n/m}$ by drawing $10^6$ pairs of $\boldsymbol{x}$ and $\boldsymbol{r}$ from their respective distributions and then computing the average of $|\boldsymbol{r}^\top \boldsymbol{x}|_1/\sqrt{n/m}$ as the estimate. The parameters of the distributions of $\boldsymbol{x}$ and $\boldsymbol{r}$ are set as follows: $m = 1$, $n = 10000$, $\mu = 1$, $\sigma = 1/3$ and $p = 1/2$ or $2/3$. The data dimension $n = 10000$ allows $k$ to vary between 1 and 10000. The average value of $|\boldsymbol{r}^\top \boldsymbol{x}|/\sqrt{n/m}$ at each $k$ is presented in Figs. 3(c) and (d), with $p = 1/2$ and $2/3$, respectively. Comparing the numerical analysis results and the simulation results shown in Fig. 3, specifically contrasting (a) vs. (c) and (b) vs. (d), it can be seen that two kinds of results are roughly consistent with each other. The consistency validates Theorem 2, as well as the numerical analysis results P5 and P6.

## 5 Experiments

In this section, we aim to verify that the impact of the varying matrix sparsity $k$ on classification is consistent with its impact on the $\ell_1$ distance between projected data as analyzed in Theorems 1 and 2; and more precisely, our goal is to demonstrate that the sparse matrices with only one or at most dozens of nonzero entries per row can provide comparable or even better classification performance than other more dense matrices, under the constraint of matrix size $m \geq \mathcal{O}(\sqrt{n})$.

### 5.1 Data

For the sake of generality, we evaluate four different types of data, including the YaleB image dataset (Georghiades et al., 2001; Lee et al., 2005), the Newsgroups text dataset (Joachims, 1997), the AMLALL gene dataset (Golub et al., 1999) and the MNIST binary image dataset (Deng, 2012). The first three kinds of data can be modeled by Gaussian mixtures, whereas the last one belongs to the two-point distribution. The data settings are introduced as follows. YaleB contains $168 \times 192$-sized face images of 38 persons, with about 64 faces per person. For easier simulation, we reduce the image size to $40 \times 30$. Newsgroups consists of 20 categories of text data, with 500 samples per category. Each sample is represented with a 3060-dimensional bag-of-words feature vector. AMLALL contains 25 samples taken from patients suffering from acute myeloid

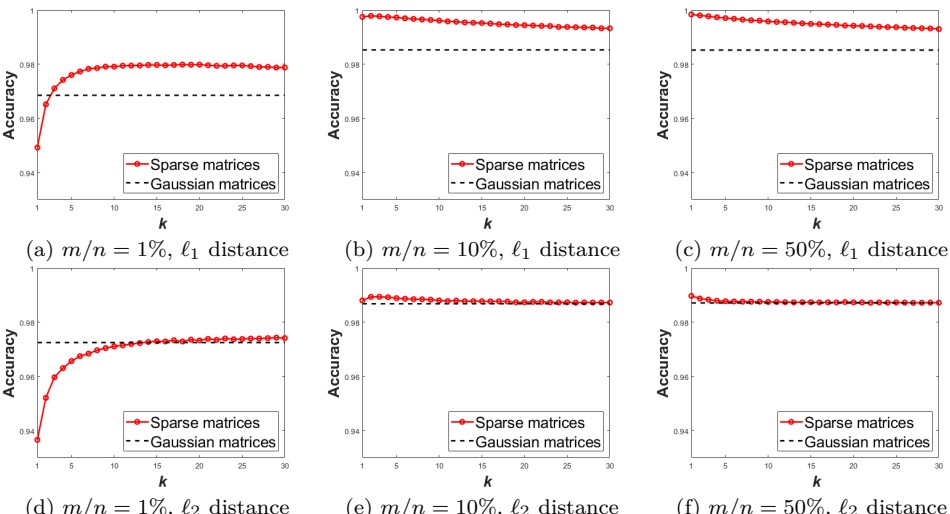

Figure 4: Classification accuracy of the sparse matrix-based and Gaussian matrices-based random projections for image data (YaleB, DCT features), with varying matrix sparsity $k \in [1, 30]$, three different projection ratios $m/n = 1\%$, 10% and 50%, and two distance metrics $\ell_1$ and $\ell_2$.

leukemia (AML) and 47 samples from patients suffering from acute lymphoblastic leukemia (ALL), with each sample expressed with a 7129-dimension gene vector. MNIST involves 10 classes of $28 \times 28$-sized handwritten digit images in MNIST, with 6000 samples per class and with each image pixel 0-1 binarized.

## 5.2 Implementation

The random projection based classification is implemented by first multiplying original data with $k$-sparse random matrices and then classifying the resulting projections with a classifier. To faithfully reflect the impact of the varying data distance on classification, we adopt the simple nearest neighbor classifier (NNC) (Cover & Hart, 1967) for classification, which has performance absolutely dependent on the pairwise distance between data points, without involving extra operations to improve data discrimination. In fact, our classification performance analysis on matrix sparsity could also be verified with other more sophisticated classifiers, like SVMs (Cortes & Vapnik, 1995), see Appendix B.

For each dataset, we will enumerate all possible class pairs in it to perform binary classification. In each class, we have one half of samples randomly selected for training and the rest for testing. To suppress the instability of random matrices and obtain relatively stable classification performance, as in (Bingham & Mannila, 2001), we repeat the random projection-based classification 5 times for each sample and make the final classification decision by voting. For comparison, we also test the performance of random projection based on Gaussian matrices, where the matrix elements are i.i.d drawn from $\mathcal{N}(0, 1)$. Additionally, the performance is also examined using $\ell_2$ distance, although our theoretical analysis is grounded on $\ell_1$ distance.

## 5.3 Results

The classification results of four kinds of data are provided in Figs. 4–7, respectively. For each kind of data, as can be seen, we evaluate the classification performance of sparse matrices with varying sparsity $k \in [1, 30]$, three different projection ratios $m/n = 1\%$, 10% and 50%, and two distance metrics $\ell_1$ and $\ell_2$. Note that the data dimensions $n$ we test here are on the order of thousands. With such scale of $n$, it is easy to deduce that the condition of $m \geq \mathcal{O}(\sqrt{n})$ will be satisfied as $m/n = 10\%$ and 50%, but be violated as $m/n = 1\%$.

Let us first examine the case of the matrices having size $m \geq \mathcal{O}(\sqrt{n})$, specifically the cases of $m/n = 10\%$ and 50% as shown in Figs. 4–7(b)(c). It is seen that the four kinds of data all achieve their best performance with relatively small matrix sparsity $k$ ($< 30$), such as with $k = 1$ in Fig. 4(c) and $k = 15$ in Fig. 5(c). But in the case of $m/n = 1\%$ which violates the condition of $m \geq \mathcal{O}(\sqrt{n})$, as shown in Figs. 4–7(a), the four kinds

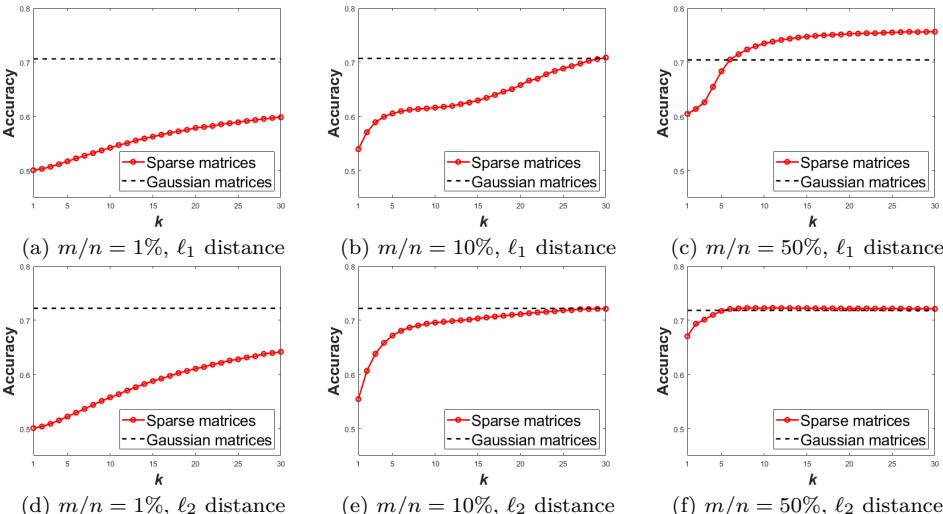

Figure 5: Classification accuracy of the sparse matrix-based and Gaussian matrix-based random projections for text data (Newsgroups), with varying matrix sparsity $k \in [1, 30]$, three different projection ratios $m/n = 1\%$, $10\%$ and $50\%$, and two distance metrics $\ell_1$ and $\ell_2$.

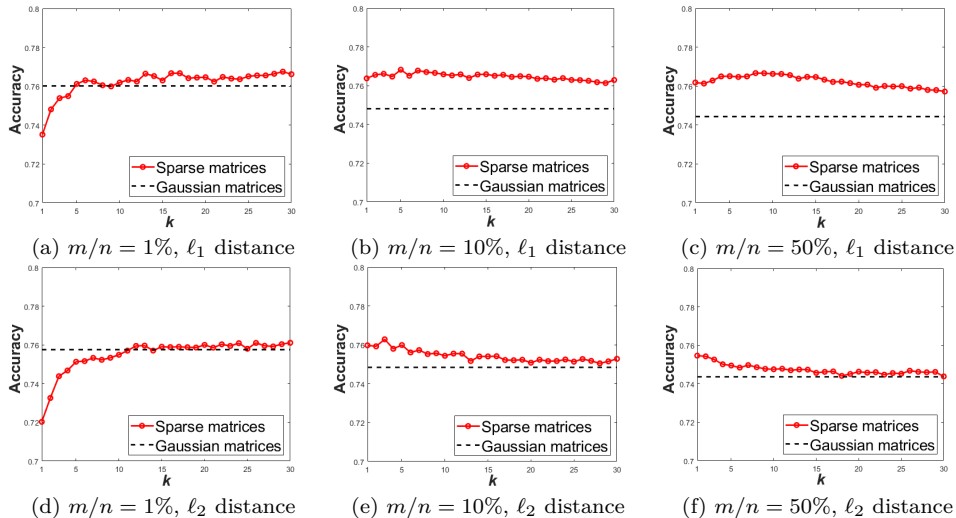

Figure 6: Classification accuracy of sparse matrix-based and Gaussian matrix-based random projections for gene data (AM-LALL), with varying matrix sparsity $k \in [1, 30]$, three different projection ratios $m/n = 1\%$, $10\%$ and $50\%$, and two distance metrics $\ell_1$ and $\ell_2$.

of data with an exception of AMLALL all fail to reach their top performance within $k < 30$. For AMLALL with $m/n = 1\%$, as illustrated in Fig. 6(a), it fails to get the desired decreasing performance trend and performs poorly at $k = 1$, in contrast to the cases of $m/n = 10\%$ and $m/n = 50\%$ shown in Figs. 6(b)(c). Overall, the experimental results on four different kinds of data all agree with our theoretical analysis: the sparse matrices with only one or at most about dozens of nonzero entries per row, achieve comparable or even better classification performance than other more dense matrices, under the size of $m \geq \mathcal{O}(\sqrt{n})$.

The trend in classification performance as the matrix sparsity $k$ varies also aligns with our theoretical analysis. More precisely, it can be seen from Figs. 4–7(b)(c) that the classifications of four datasets quickly converge to stable performance with the increasing matrix sparsity $k$. The difference between them mainly lies in the initial stage of the convergence. Specifically, as illustrated in Figs. 4(b)(c) and 6(b)(c), the convergence curves on the datasets YaleB and AMLALL both exhibit the declining trend at the initial increasing region of $k$, consistent with the numerical analysis result depicted in Fig. 3(a) (discussed in P5 and P6). As for the

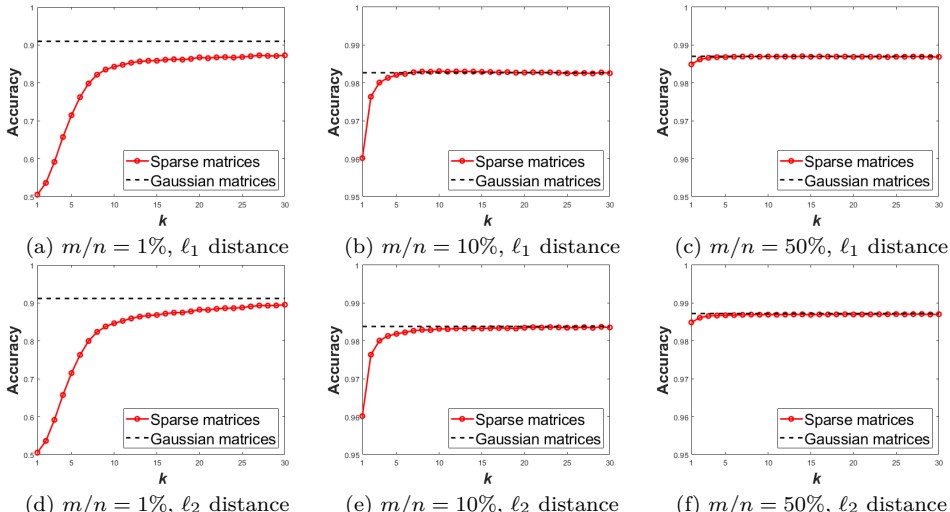

Figure 7: Classification accuracy of the sparse matrix-based and Gaussian matrix-based random projections for binary image data (MNIST, binarized pixels), with varying matrix sparsity $k \in [1, 30]$, three different projection ratios $m/n = 1\%$, 10% and 50%, and two distance metrics $\ell_1$ and $\ell_2$.

curves on the other two datasets Newsgroups and MNIST, as shown in Figs. 5(b)(c) and Figs. 7(b)(c), they both exhibit the trend of initially increasing with $k$, aligning with the numerical analysis results illustrated in Fig. 3(b) (P5 and P6) and Fig. 1(b) (P4 and P5).

Although the classification performance of sparse matrices is analyzed with $\ell_1$ distance, it can be seen that the performance also holds for $\ell_2$ distance, when comparing the upper row and the bottom row results shown in Figs. 4–7. This generalization can be attributed to the closeness of the two metrics (Gionis et al., 1999; Figiel et al., 1977). Moreover, experiments show that sparse matrices perform comparably or even better than the popularly used Gaussian matrices. This encourages us to replace Gaussian matrices with sparse matrices, for their much lower complexity.

## 6 Conclusion

For the sparse $\{0, \pm 1\}$-matrix-based random projection, we analyzed the influence of matrix sparsity on classification. We found that sparse matrices with one or at most dozens of nonzero entries per row can offer comparable or superior classification performance compared to denser matrices, especially when the matrix size is $m \geq \mathcal{O}(\sqrt{n})$ and the original data are highly discriminative. Empirically, these sparse matrices also outperform popular Gaussian matrices. Furthermore, the performance advantages observed with $\ell_1$ distance also apply to $\ell_2$ distance. These results suggest broad applicability for our sparse matrices. Notably, our theoretical analysis aligns well with experiments on various real-world datasets, which can be attributed to the strong generalizability of the two data models used in our statistical analysis.

Besides the contribution to random projection, our analysis of classification performance using sparse matrices also sheds light on the competitive capabilities of deep ternary networks. These networks are formed by ternarizing the parameters and/or activations of full-precision networks, resulting in highly sparse architectures (Li et al., 2016; Zhu et al., 2017; Wan et al., 2018; Marban et al., 2020; Rokh et al., 2023). Despite significant quantization errors, ternary networks often experience acceptable performance degradation and, surprisingly, sometimes even outperform their full-precision counterparts. The underlying cause of this intriguing behavior remains unknown. Given that deep networks can be conceptualized as a sequence of random projections (Giryes et al., 2016), our analysis of sparse matrix-based random projection can be seen as a layer-by-layer examination of deep ternary networks. The sparse ternary matrices we have identified with robust classification performance partially explain the strong performance of sparse ternary networks.

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

# A  Appendix

## A.1  Proof of Theorem 1

*Proof.* In the following, we sequentially prove (4), (5), P1 and P2.

**Proofs of** (4) **and** (5)**:** With the distributions of $\boldsymbol{r}$ and $\boldsymbol{x}$, we can write $\|\boldsymbol{r}^\top\boldsymbol{x}\|_1 = \sqrt{\frac{n}{mk}}\mu\left|\sum_{i=1}^k z_i\right|$, where $z_i \in \{-1,0,1\}$ with probabilities $\{q,p,q\}$. Then, it can be derived that

$$\mathbb{E}|\boldsymbol{r}^\top\boldsymbol{x}| = \mu\sqrt{\frac{n}{mk}}\sum_{i=0}^k C_k^i p^i q^{k-i}\sum_{j=0}^{k-i} C_{k-i}^j |k-i-2j|, \tag{15}$$

among which $\sum_{j=0}^{k-i} C_{k-i}^j |k-i-2j|$ can be expressed as

$$\sum_{j=0}^{k-i}(C_{k-i}^j |k-i-2j|) = 2\left\lceil\frac{k-i}{2}\right\rceil C_{k-i}^{\lceil\frac{k-i}{2}\rceil}, \tag{16}$$

where $\lceil\alpha\rceil = \min\{\beta : \beta \geq \alpha, \beta \in \mathbb{Z}\}$. Combine (15) and (16), we can obtain (4).

Next, we can derive the variance of $|\boldsymbol{r}^\top\boldsymbol{x}|$

$$\begin{aligned}\text{Var}(|\boldsymbol{r}^\top\boldsymbol{x}|) &= \text{Var}(\boldsymbol{r}^\top\boldsymbol{x}) - \left(\mathbb{E}|\boldsymbol{r}^\top\boldsymbol{x}|\right)^2 \\ &= \frac{2q\mu^2 n}{m} - \frac{4\mu^2 n}{mk}\left(\sum_{i=0}^k C_k^i p^i q^{k-i}\left\lceil\frac{k-i}{2}\right\rceil C_{k-i}^{\lceil\frac{k-i}{2}\rceil}\right)^2.\end{aligned} \tag{17}$$

**Proof of P1:** This part aims to prove

$$\mathbb{E}|\boldsymbol{r}^\top\boldsymbol{x}|_{k=1} > \mathbb{E}|\boldsymbol{r}^\top\boldsymbol{x}|_{k>1},$$

where the subscript $k=1$ denotes the case of $\mathbb{E}|\boldsymbol{r}^\top\boldsymbol{x}|$ with $k=1$, and the subscript $k>1$ means the case of $k$ taking any integer value greater than 1. In the following, we will calculate and compare $\mathbb{E}|\boldsymbol{r}^\top\boldsymbol{x}|$ in terms of the two cases. For the case of $k=1$, by (4), it is easy to derive that

$$\mathbb{E}|\boldsymbol{r}^\top\boldsymbol{x}|_{k=1} = 2q\mu\sqrt{\frac{n}{m}}. \tag{18}$$

Then, let us see the case of computing $\mathbb{E}|\boldsymbol{r}^\top\boldsymbol{x}|_{k>1}$. By (4), $\mathbb{E}|\boldsymbol{r}^\top\boldsymbol{x}|_{k>1}$ is the sum of $\frac{2}{\sqrt{k}}C_k^i p^i q^{k-i}\left\lceil\frac{k-i}{2}\right\rceil C_{k-i}^{\lceil\frac{k-i}{2}\rceil}$ multiplied by $\mu\sqrt{\frac{n}{m}}$. To compute $\frac{2}{\sqrt{k}}C_k^i p^i q^{k-i}\left\lceil\frac{k-i}{2}\right\rceil C_{k-i}^{\lceil\frac{k-i}{2}\rceil}$, we consider separately two cases: $k-i$ is even or odd, as detailed below.

**Case 1:** Suppose $k-i$ is even. We have

$$\begin{aligned}&\frac{2}{\sqrt{k}}C_k^i p^i q^{k-i}\left\lceil\frac{k-i}{2}\right\rceil C_{k-i}^{\lceil\frac{k-i}{2}\rceil} \\ &\leq \frac{1}{\sqrt{k}}C_k^i p^i q^{k-i}(k-i)2^{k-i}\sqrt{\frac{2}{(k-i)\pi}} \\ &\leq \sqrt{\frac{2}{\pi}}C_k^i p^i (2q)^{k-i},\end{aligned} \tag{19}$$

since $C_{2\gamma}^\gamma \leq \frac{2^{2\gamma}}{\sqrt{\gamma\pi}}$, where $\gamma$ is a positive integer (Stănică, 2001).

**Case 2:** Suppose $k - i$ is odd. We have

$$\frac{2}{\sqrt{k}} C_k^i p^i q^{k-i} \left\lceil \frac{k-i}{2} \right\rceil C_{k-i}^{\lceil \frac{k-i}{2} \rceil}$$

$$\leq \frac{1}{\sqrt{k}} C_k^i p^i q^{k-i} (k-i) 2^{k-i} \sqrt{\frac{2}{(k-i-1)\pi}}$$

$$= \sqrt{\frac{2}{\pi}} C_k^i p^i (2q)^{k-i} \frac{k-i}{\sqrt{k(k-i-1)}} \tag{20}$$

Given $k \geq 5$, we further have

$$\frac{k-i}{\sqrt{k(k-i-1)}} < 1 \quad \text{for} \ \ 2 \leq i \leq k-2,$$

and for $i = k-1$ or $k$,

$$\frac{2}{\sqrt{k}} C_k^i p^i q^{k-i} \left\lceil \frac{k-i}{2} \right\rceil C_{k-i}^{\lceil \frac{k-i}{2} \rceil} < \sqrt{\frac{2}{\pi}} C_k^i p^i (2q)^{k-i}.$$

To sum up, when $k - i$ is odd,

$$\frac{2}{\sqrt{k}} C_k^i p^i q^{k-i} \left\lceil \frac{k-i}{2} \right\rceil C_{k-i}^{\lceil \frac{k-i}{2} \rceil}$$

$$\leq \begin{cases} \sqrt{\dfrac{2}{\pi}} C_k^i p^i (2q)^{k-i}, & k \geq 5, i \geq 2, \\ \dfrac{2}{\sqrt{k}} C_k^i p^i q^{k-i} (k-i) C_{k-i-1}^{\frac{k-i-1}{2}}, & \text{otherwise.} \end{cases} \tag{21}$$

According to the results (19) and (21) derived in the above two cases, we know that $\mathbb{E}|\boldsymbol{r}^\top \boldsymbol{x}|_{k>1}$ can be computed in terms of two cases, $2 \leq k \leq 4$ and $k \geq 5$. For the case of $2 \leq k \leq 4$, by (4), we have

$$\mathbb{E}|\boldsymbol{r}^\top \boldsymbol{x}| = \begin{cases} \dfrac{\mu \sqrt{n}}{\sqrt{2m}} (4q^2 + 4pq), & k = 2, \\ \dfrac{\mu \sqrt{n}}{\sqrt{3m}} (12q^3 + 12pq^2 + 6p^2 q), & k = 3, \\ \dfrac{\mu \sqrt{n}}{\sqrt{m}} (12q^4 + 24pq^3 + 12p^2 q^2 + 4p^3 q), & k = 4, \end{cases} \tag{22}$$

and for the case of $k \geq 5$, with (19) and (21), we have

$$\mathbb{E}|\boldsymbol{r}^\top \boldsymbol{x}| \leq \mu \sqrt{\frac{2n}{\pi m}} + \mu \sqrt{\frac{n}{m}} (2q)^5 \left( \frac{3\sqrt{5}}{8} - \sqrt{\frac{2}{\pi}} \right). \tag{23}$$

By (18), (22) and (23), we can derive that

$$\mathbb{E}|\boldsymbol{r}^\top \boldsymbol{x}|_{k=1} > \mathbb{E}|\boldsymbol{r}^\top \boldsymbol{x}|_{k>1}$$

holds under the condition of $p \leq 0.188$. Then P1 is proved.

In what follows, we elaborate the proof of (23) by considering two cases of $k$, being even or odd.

**Case 1:** Suppose $k \geq 5$ and $k$ is even. Combining (19) and (21), we have

$$\mathbb{E}|\boldsymbol{r}^\top \boldsymbol{x}| \leq \mu\sqrt{\frac{n}{m}}C_k^1 p(2q)^{k-1}\left(\frac{\sqrt{k}}{2^{k-1}}C_{k-1}^{\frac{k}{2}-1} - \sqrt{\frac{2}{\pi}}\right)$$
$$+ \mu\sqrt{\frac{2n}{\pi m}}\sum_{i=0}^{k}C_k^i p^i(2q)^{k-i}. \tag{24}$$

Denote $h_1(k) = \frac{\sqrt{k}}{2^{k-1}}C_{k-1}^{\frac{k}{2}-1}$. For

$$\frac{h_1(k+2)}{h_1(k)} = \frac{k+1}{\sqrt{k(k+2)}} > 1$$

we have

$$h_1(k) = \frac{\sqrt{k}}{2^{k-1}}C_{k-1}^{\frac{k}{2}-1} \leq \lim_{k\to\infty}h_1(k) = \sqrt{\frac{2}{\pi}}. \tag{25}$$

Then, it follows from (24) and (25) that

$$\mathbb{E}|\boldsymbol{r}^\top \boldsymbol{x}| \leq \mu\sqrt{\frac{2n}{\pi m}}. \tag{26}$$

**Case 2:** Suppose $k \geq 5$ and $k$ is odd. Combining (19) and (21), we have

$$\mathbb{E}|\boldsymbol{r}^\top \boldsymbol{x}| \leq \mu\sqrt{\frac{n}{m}}C_k^0(2q)^k\left(\frac{\sqrt{k}}{2^{k-1}}C_{k-1}^{\frac{k-1}{2}} - \sqrt{\frac{2}{\pi}}\right)$$
$$+ \mu\sqrt{\frac{2n}{\pi m}}\sum_{i=0}^{k}C_k^i p^i(2q)^{k-i}. \tag{27}$$

Denote $h_2(k) = \frac{\sqrt{k}}{2^{k-1}}C_{k-1}^{\frac{k-1}{2}}$. For

$$\frac{h_2(k+2)}{h_2(k)} = \frac{\sqrt{k(k+2)}}{k+1} < 1$$

we have

$$h_2(k) = \frac{\sqrt{k}}{2^{k-1}}C_{k-1}^{\frac{k-1}{2}} \leq h_2(5) = \frac{\sqrt{5}}{2^4}C_4^2. \tag{28}$$

Then, it follows from (27) and (28) that

$$\mathbb{E}|\boldsymbol{r}^\top \boldsymbol{x}| \leq \mu\sqrt{\frac{2n}{\pi m}} + \mu\sqrt{\frac{n}{m}}(2q)^5\left(\frac{3\sqrt{5}}{8} - \sqrt{\frac{2}{\pi}}\right).$$

**Proof of P2:** For ease of analysis, we first define the function

$$g(\boldsymbol{r}^\top \boldsymbol{x}; k, p) = \frac{\mathbb{E}|\boldsymbol{r}^\top \boldsymbol{x}|_k}{\mu\sqrt{n/m}} = \mathbb{E}\left|\frac{1}{\sqrt{k}}\sum_{i=1}^{k}z_i\right|, \tag{29}$$

where $\{z_i\}$ is independently and identically distributed and $z_i \in \{-1, 0, 1\}$ with probabilities $\{q, p, q\}$. By the Lindeberg-Lévy Central Limit Theorem, we have

$$\frac{1}{\sqrt{k}}\sum_{i=1}^{k}z_i \rightsquigarrow Z, \tag{30}$$

where $Z \sim N(0, 2q)$.

Then based on (23), we have for $k \geq 5$,

$$\mathbb{E}\left|\frac{1}{\sqrt{k}}\sum_{i=1}^{k} z_i\right| \leq \sqrt{\frac{2}{\pi}} + (2q)^5 \left(\frac{3\sqrt{5}}{8} - \sqrt{\frac{2}{\pi}}\right).$$

It means that

$$\lim_{M \to +\infty} \limsup_{k \to +\infty} \mathbb{E}\left[\left|\frac{1}{\sqrt{k}}\sum_{i=1}^{k} z_i\right| 1\left\{\left|\frac{1}{\sqrt{k}}\sum_{i=1}^{k} z_i\right| > M\right\}\right] = 0.$$

Hence, $\left|\frac{1}{\sqrt{k}}\sum_{i=1}^{k} z_i\right|$ is an asymptotically uniformly integrable sequence.

According to Theorem 2.20 in (Van der Vaart, 2000), we obtain

$$\lim_{k \to +\infty} \frac{\sqrt{m}}{\mu\sqrt{n}} \mathbb{E}|\boldsymbol{r}^\top \boldsymbol{x}| = \lim_{k \to +\infty} \mathbb{E}\left|\frac{1}{\sqrt{k}}\sum_{i=1}^{k} z_i\right|$$

$$= \mathbb{E}|Z|$$

$$= 2\sqrt{\frac{q}{\pi}}.$$

Next, let us investigate the error of the above convergence with respect to $k$. Following the definitions and properties described in Eqs. (29) and (30), we further suppose $t_i = \frac{1}{\sqrt{2q}} z_i$ and $Q \sim N(0, 1)$, and get

$$\left|\frac{\sqrt{m}}{\mu\sqrt{n}} \mathbb{E}|\boldsymbol{r}^\top \boldsymbol{x}| - 2\sqrt{q/\pi}\right|$$

$$= \left|\mathbb{E}\left|\frac{1}{k}\sum_{i=1}^{k} z_i\right| - \mathbb{E}|Z|\right|$$

$$= \sqrt{2q}\left|\mathbb{E}\left|\frac{1}{k}\sum_{i=1}^{k} t_i\right| - \mathbb{E}|Q|\right|$$

$$\leq \sqrt{2q} d_w \left(\mathbb{E}\left|\frac{1}{k}\sum_{i=1}^{k} t_i\right|, \mathbb{E}|Q|\right)$$

where $d_w(\nu, \upsilon)$ denotes the Kolmogorov metric, with the form

$$d_w(\nu, \upsilon) = \sup_{h \in \mathcal{H}}\left|\int h(x)d\nu(x) - \int h(x)d\upsilon(x)\right|,$$
$$\mathcal{H} = \{h : \mathbb{R} \to \mathbb{R} : |h(x) - h(y)| \leq |x - y|\}.$$

By the Theorem 3.2 in Ross (2011), since $\{t_i\}$ are i.i.d and $\mathbb{E}t_i = 0$, $\mathbb{E}t_i^2 = 1$, $\mathbb{E}|t_i|^4 < \infty$, we have

$$d_w\left(\mathbb{E}\left|\frac{1}{k}\sum_{i=1}^{k} t_i\right|, \mathbb{E}|Q|\right) \leq \frac{1}{k^{3/2}}\sum_{i=1}^{k}\mathbb{E}|t_i|^3 + \frac{\sqrt{2}}{\sqrt{\pi}k}\sqrt{\sum_{i=1}^{k}\mathbb{E}t_i^4}$$

$$= \frac{1}{\sqrt{2qk}} + \frac{\sqrt{2}}{\sqrt{2q\pi k}},$$

and then

$$\left|\frac{\sqrt{m}}{\mu\sqrt{n}} \mathbb{E}|\boldsymbol{r}^\top \boldsymbol{x}| - 2\sqrt{q/\pi}\right| \leq \frac{\sqrt{\pi} + \sqrt{2}}{\sqrt{\pi k}}.$$

$\square$

### A.2 Proof of Property 1

*Proof.* This problem can be addressed using the Chebyshev's Inequality, which requires us to first derive $\mathbb{E}z$ and $\text{Var}(z)$. Note that $\mathbb{E}z = \mathbb{E}(\frac{1}{m}\sum_{i=1}^{m}|\boldsymbol{r}_i^\top\boldsymbol{x}|) = \mathbb{E}(|\boldsymbol{r}_i^\top\boldsymbol{x}|)$ has been derived in (4). In the sequel, we need to first solve $\text{Var}(z) = \mathbb{E}z^2 - (\mathbb{E}z)^2$, which has

$$
\begin{aligned}
\mathbb{E}z^2 &= \mathbb{E}(\frac{1}{m}\sum_{i=1}^{m}|\boldsymbol{r}_i^\top\boldsymbol{x}|)^2 \\
&= \frac{1}{m^2}\mathbb{E}(\sum_{i=1}^{m}|\boldsymbol{r}_i^\top\boldsymbol{x}|^2) + \frac{1}{m^2}\mathbb{E}(\sum_{i\neq j}|\boldsymbol{r}_i^\top\boldsymbol{x}|\cdot|\boldsymbol{r}_j^\top\boldsymbol{x}|) \\
&= \frac{2q\mu^2 n}{m^2} + \frac{m-1}{2m}\mathbb{E}(|\boldsymbol{r}_i^\top\boldsymbol{x}|\cdot|\boldsymbol{r}_j^\top\boldsymbol{x}|).
\end{aligned}
\tag{31}
$$

For the second term in the above result, it holds

$$
\mathbb{E}(|\boldsymbol{r}_i^\top\boldsymbol{x}|\cdot|\boldsymbol{r}_j^\top\boldsymbol{x}|) \leq \text{Var}(|\boldsymbol{r}_i^\top\boldsymbol{x}|) + (\mathbb{E}|\boldsymbol{r}_i^\top\boldsymbol{x}|)^2 = \text{Var}(|\boldsymbol{r}_i^\top\boldsymbol{x}|) + (\mathbb{E}z)^2,
\tag{32}
$$

by the covariance property

$$
\begin{aligned}
\text{Cov}(|\boldsymbol{r}_i^\top\boldsymbol{x}|,|\boldsymbol{r}_j^\top\boldsymbol{x}|) &= \mathbb{E}(|\boldsymbol{r}_i^\top\boldsymbol{x}|\cdot|\boldsymbol{r}_j^\top\boldsymbol{x}|) - \mathbb{E}|\boldsymbol{r}_i^\top\boldsymbol{x}|\cdot\mathbb{E}|\boldsymbol{r}_j^\top\boldsymbol{x}| \\
&= \rho\sqrt{\text{Var}(|\boldsymbol{r}_i^\top\boldsymbol{x}|)}\cdot\sqrt{\text{Var}(|\boldsymbol{r}_j^\top\boldsymbol{x}|)} \\
&= \rho\text{Var}(|\boldsymbol{r}_i^\top\boldsymbol{x}|),
\end{aligned}
\tag{33}
$$

where $\rho \in (-1,1)$ is the correlation coefficient.

Substituting (31) into $\text{Var}(z) = \mathbb{E}z^2 - (\mathbb{E}z)^2$, by the inequality (32) and (17), we can derive

$$
\begin{aligned}
\text{Var}(z) &\leq \frac{2q\mu^2 n}{m^2} + \frac{m-1}{2m}[\text{Var}(|\boldsymbol{r}_i^\top\boldsymbol{x}|) + (\mathbb{E}z)^2] - (\mathbb{E}z)^2 \\
&= \frac{2q\mu^2 n}{m^2} + \frac{m-1}{2m}\cdot\frac{2q\mu^2 n}{m^2} - (\mathbb{E}z)^2 \\
&= \frac{(m+1)q\mu^2 n}{m^2} - (\mathbb{E}z)^2.
\end{aligned}
\tag{34}
$$

With the above inequality about $\text{Var}(z)$, we can further explore the condition that holds the desired probability

$$
\Pr\{|z - \mathbb{E}z| \leq \varepsilon\} \geq 1 - \delta.
\tag{35}
$$

By the Chebyshev's Inequality, (35) will be achieved, if $\text{Var}(z)/\varepsilon^2 \leq \delta$; and according to (34), this condition can be satisfied when $\frac{m^2}{m+1} \geq \frac{q\mu^2 n}{\varepsilon^2\delta}$.

In the above analysis, we consider a random $\boldsymbol{x}$. For a given $\boldsymbol{x}$, the condition of holding (35) can be further relaxed to $m^2 \geq \frac{2q\mu^2 n}{\varepsilon^2\delta}$, since in this case $|\boldsymbol{r}_i^\top\boldsymbol{x}|$ is independent between different $i \in [m]$, such that $\text{Var}(z)$ changes to be (17) divided by $m$. $\qquad\square$

### A.3 Proof of Theorem 2

*Proof.* First, we derive the absolute moment of $z \sim \mathcal{N}(\mu, \sigma^2)$ as

$$
\mathbb{E}|z| = \sqrt{\frac{2}{\pi}}\sigma e^{-\frac{\mu^2}{2\sigma^2}} + \mu\left(1 - 2\Phi\left(-\frac{\mu}{\sigma}\right)\right)
\tag{36}
$$

which will be used in the sequel. With the distributions of $\boldsymbol{r}$ and $\boldsymbol{x}$, we have $|\boldsymbol{r}^\top \boldsymbol{x}| = \sqrt{\frac{n}{mk}} \left| \sum_{i=1}^{k} x_i \right|$. For easier expression, assume $y = \sum_{i=1}^{k} x_i$, then the distribution of $y$ can be expressed as

$$f(y) = \sum_{i=0}^{k} \sum_{j=0}^{k-i} C_k^i C_{k-i}^j p^i q^{k-i} \frac{1}{\sqrt{2\pi k}\sigma} e^{-\frac{(y-(2j+i-s)\mu)^2}{2k\sigma^2}}.$$

Then, by (36) we can derive that

$$
\begin{aligned}
\mathbb{E}|\boldsymbol{r}^\top \boldsymbol{x}| = {} & \sqrt{\frac{n}{mk}} \sum_{i=0}^{k} \sum_{j=0}^{k-i} \Big[ C_k^i C_{k-i}^j p^i q^{k-i} \\
& \times \int_{-\infty}^{+\infty} \frac{|y|}{\sqrt{2\pi k}\sigma} e^{-\frac{(y-(2j+i-s)\mu)^2}{2k\sigma^2}} dy \Big] \\
= {} & 2\mu \sqrt{\frac{n}{mk}} \sum_{i=0}^{k} C_k^i p^i q^{k-i} \left\lceil \frac{k-i}{2} \right\rceil C_{k-i}^{\lceil \frac{k-i}{2} \rceil} \\
& - 2\mu \sqrt{\frac{n}{mk}} \sum_{i=0}^{k} C_k^i p^i q^{k-i} \sum_{j=0}^{k-i} C_{k-i}^j \Phi\left( -\frac{|k-i-2j|\mu}{\sqrt{k}\sigma} \right) \\
& + \sigma \sqrt{\frac{2n}{\pi m}} \sum_{i=0}^{k} C_k^i p^i q^{k-i} \sum_{j=0}^{k-i} C_{k-i}^j e^{-\frac{(k-i-2j)^2 \mu^2}{2k\sigma^2}}
\end{aligned}
$$

where $\Phi(\cdot)$ is the distribution function of $\mathcal{N}(0,1)$.

The above equation and (18), (22), (23) together lead to

$$\mathbb{E}|\boldsymbol{r}^\top \boldsymbol{x}| \leq \mu \sqrt{\frac{n}{m}} + \sigma \sqrt{\frac{2n}{\pi m}}.$$

Next, we can derive the variance of $|\boldsymbol{r}^\top \boldsymbol{x}|$ as

$$
\begin{aligned}
\mathrm{Var}(|\boldsymbol{r}^\top \boldsymbol{x}|) &= Var(\boldsymbol{r}^\top \boldsymbol{x}) - \left( \mathbb{E}|\boldsymbol{r}^\top \boldsymbol{x}| \right)^2 \\
&= \frac{n}{m}(\sigma^2 + 2q\mu^2) - \left( \mathbb{E}\|\boldsymbol{r}^\top \boldsymbol{x}\|_1 \right)^2.
\end{aligned}
$$

Finally, the convergence of $\frac{\sqrt{m}}{\mu\sqrt{n}} \mathbb{E}|\boldsymbol{r}^\top \boldsymbol{x}|$ shown in (12) and (13) can be derived in a similar way to the proof of P2 in Theorem 1. □

## B  Appendix

In Figs. 8–11, we test the SVM (with linear kernel) classification accuracy for the sparse ternary matrix with varying matrix sparsity $k$ (and compression ratio $m/n$) on four different types of data. It can be seen that the performance changing trends of SVM against the varying matrix sparsity $k$ are similar to the KNN performance as illustrated in the body of the paper, thus consistent with our theoretical analysis.

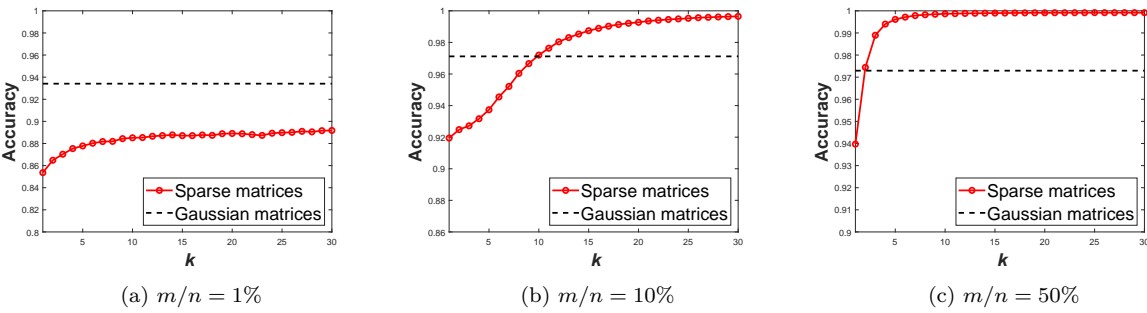

Figure 8: Classification accuracy of the sparse matrix-based and Gaussian matrix-based random projections for image data (YaleB, DCT features), with varying matrix sparsity $k \in [1, 30]$, three different projection ratios $m/n = 1\%$, 10% and 50%.

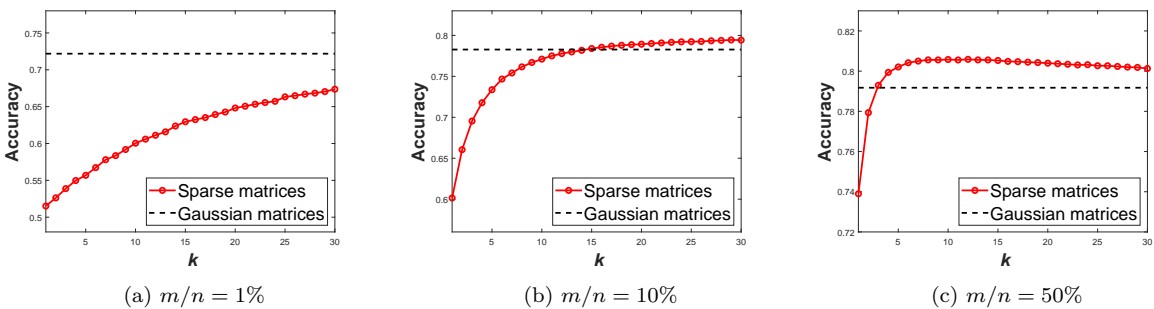

Figure 9: Classification accuracy of the sparse matrix-based and Gaussian matrix-based random projections for text data (Newsgroups), with varying matrix sparsity $k \in [1, 30]$, three different projection ratios $m/n = 1\%$, 10% and 50%.

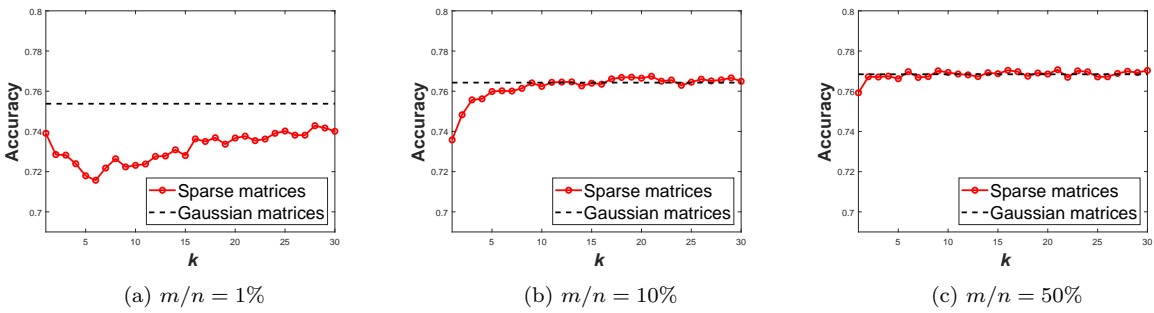

Figure 10: Classification accuracy of the sparse matrix-based and Gaussian matrix-based random projections for gene data (AMLALL), with varying matrix sparsity $k \in [1, 30]$, three different projection ratios $m/n = 1\%$, 10% and 50%.

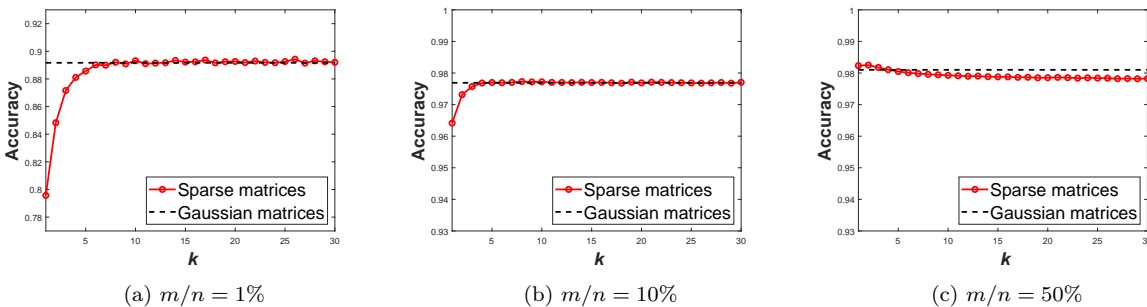

Figure 11: Classification accuracy of the sparse matrix-based and Gaussian matrix-based random projections for binary image data (MNIST, binarized pixels), with varying matrix sparsity $k \in [1, 30]$, three different projection ratios $m/n = 1\%$, 10% and 50%.

