# OpenReview forum: "The Sparse Matrix-Based Random Projection: An Analysis of Matrix Sparsity for Classification"
_TMLR — Rejected by TMLR_

### Review · Reviewer_ViCE · 2024-09-17

**Summary Of Contributions:**

In this paper, the authors utilize the $l_1$ norm (also known as the distance) and subsequently investigate sparse matrix-based random projection, with an analysis conducted under two data distribution assumptions: the Gaussian mixture distribution and the two-point distribution.

**Audience:**

Yes

**Claims And Evidence:**

Yes

**Requested Changes:**

I have reviewed the additional theorem in the new version and believe it addresses the concerns I previously raised.

**Strengths And Weaknesses:**

The paper presents the performance benefits of utilizing sparse matrix-based random projection with the $l_1$  distance metric.

---

### Review · Reviewer_gLeX · 2025-01-03

**Summary Of Contributions:**

The paper studies the effect of sparsity $k$ on the $\ell_1$ distance between projected data points for two different data distributions, namely Gaussian mixture model and two-point distribution. It explores the minimum number of non-zero entires $\pm1$ that supports achieving the nearly best classification performance on the projected data.

**Audience:**

Yes

**Broader Impact Concerns:**

I do not foresee any concerns on the ethical implications of the work

**Claims And Evidence:**

Yes

**Requested Changes:**

I would request the authors to comment on the points on weakness mentioned in the section above. Further what are $p$ and $q$ in equation (2)? Are the results generalisable to Guassian mixture models with $\geq 2$ mixture components?

**Strengths And Weaknesses:**

Strengths:
1. The paper is very well written and easy to read.
2. Studies the novel  problem, namely the impact of sparsity on classification performance.
3. Theoretical results are quite interesting
4. Experimental results are presented to corroborate theoretical findings.

Weakness:
1. The authors need to comment on why the $\ell_1$ distance is analysed and not say $\ell_2$?
2. It is not vey clear why maximising the $\ell_1$ distance leads to better classification performance.

---

> ### Author Response · Authors · 2025-01-07
> **Response to Reviewer gLeX**
>
> Dear Reviewer gLeX,
>
> Thank you very much for dedicating your valuable time to review our manuscript. Below we reply to the comments one by one.
>
> **C1:** The authors need to comment on why the $\ell_1$ distance is analysed and not say $\ell_2$?
>
> **R1:** Thank you. For the sparse matrix-base random projection, it has been proved that the expected distance $E ||r^\top x||$ can be preserved in $\ell_2$ norm [r1, r2], but does *not* hold in $\ell_1$ norm [r3, r4]. Therefore, we propose to investigate the impact of matrix sparsity on the varying of the expected $\ell_1$ distance. This issue has been mentioned in the introduction.
>
> [r1] D. Achlioptas. Database-friendly random projections: Johnson–Lindenstrauss with binary coins. J. Comput. Syst. Sci., 66(4):671–687, 2003.
>
> [r2] P. Li, T. J. Hastie, and K. W. Church. Very sparse random projections. in Proceedings of the 12th ACM SIGKDD international conference on Knowledge discovery and data mining, 2006.
>
> [r3] Bo Brinkman and Moses Charikar. On the impossibility of dimension reduction in $\ell_1$. Journal of the ACM, pp. 766–788, 2003.
>
> [r4] Ping Li. Very sparse stable random projections for dimension reduction in $\ell_\alpha$ ($0 < \alpha \leq2$) norm. In Proceedings of the 13th ACM SIGKDD International Conference on Knowledge Discovery and Data Mining, 2007.
>
> **C2:** It is not very clear why maximising the $\ell_1$ distance leads to better classification performance.
>
>
> **R2:** Thank you. This argument originates from principal component analysis (PCA) [r1], constituting a fundamental aspect of PCA.  In PCA, it is argued that projecting data onto  *larger* principal components results in  *larger* variances  (achievable by *larger* pairwise distances) among the projected data points; and these *larger* variances are beneficial as they tend to capture more of the variation (i.e., statistical information) inherent in the original data [r2], thereby achieving better classification performance. This principle has been widely recognized and validated in various applications of PCA, such as face recognition [r3]. We have discussed this issue in the introduction.
>
> [r1] Ian T Jolliffe. Principal component analysis. Springer, 2002.
>
> [r2] Ian T Jolliffe and Jorge Cadima. Principal component analysis: a review and recent developments. Philosophical transactions of the royal society A: Mathematical, Physical and Engineering Sciences, 374(2065): 20150202, 2016.
>
> [r3] M.A. Turk and A.P. Pentland. Face recognition using eigenfaces. In Proceedings. 1991 IEEE Computer Society Conference on Computer Vision and Pattern Recognition, pp. 586–591, 1991.
>
>  **C3:** what are $p$ and $q$ in equation (2)? Are the results generalisable to Guassian mixture models with $\geq2$ mixture components?
>
> **R3:** Thank you.  In equations (2) and (3), we define a three-component Gaussian mixture model for each element $x_i$ of the difference vector $x=h-h’$, where $h$, $h’\in \mathbb{R}^n$ are two original data points. As shown in (3), the parameters $p$, $q$, $q$ represent the mixture weights of the three components, with $p+2q=1$.
>
> Our choice of a three-component model is based on the following fact. Given two original data points $h$ and $h’$, each pair of their elements $h_i$ and $h’_i$ should be either drawn from the same distribution, or not. This implies that the difference between them, namely $x_i=h-h’$, will either have a zero-mean distribution or a nonzero mean $\mu\neq0$. Considering the sign, we further divide the nonzero-mean case into two subcases: $\mu$ and $-\mu$, $\mu>0$. Thus, we derive the three-component model as specified in (3).
>
> Given this analysis, the three-component model is a reasonable choice for modeling real data. This is supported by the consistency between our theoretical and experimental results. Therefore, there is no need to further increase the number of mixture components, as it would significantly complicate the subsequent theoretical analysis. We will include the above discussion in the revised manuscript.

---

> ### Comment · Reviewer_gLeX · 2025-01-09
> **Few more clarifications**
>
> 1. To further understand the need for $\ell_1$ analysis, is the $\ell_2$ norm of $\mathbb{E}[ \left| r^T x\right|]$ preserved irrespective of the sparsity? If not, how sparsity impacts it?
> 2. I find the argument that larger distances improves classification accuracy to be too-general and vague. As per the authors argument, $\ell_2$ distances are preserved by sparse matrix-base random projection. Does that mean the classification accuracy is the same before and after projections as far as $\ell_2$ is concerned? It would be useful if the authors could be more specific about how maximising the $\ell_1$ distance (not just any distance) leads to improved classification accuracy.
> 3. The equation (3) is unclear. I agree to its validity when each element of $h, h^{\prime}$ are samples from a univariate Gaussian distribution with mean $\mu$ and variance $\sigma^2$. Can the authors elaborate on why it still holds when each entry is itself from a Gaussian mixture model with say $M$ components?

---

> ### Author Response · Authors · 2025-01-09
> **Response to Reviewer gLeX**
>
> Dear Reviewer gLeX,
>
> Thank you for your further comments, which are explained below.
>
> **C1:**  To further understand the need for $\ell_1$ analysis, is the $\ell_2$  norm of $E|r^\top x|$  preserved irrespective of the sparsity? If not, how sparsity impacts it?
>
> **R1:**  Thank you. By [r1, r2], the $\ell_2$  norm-expectation $E||r^\top x||_2$ is indeed irrespective of the sparsity, but its variance $Var(||r^\top x||_2)$ is linked to the sparsity:  the variance $Var(||r^\top x||_2)$  tends to increase, as the sparsity $k$ decreases. This implies that, with the decreasing sparsity $k$, the probability of an *actual* matrix's  distance $||r^\top x||_2$ approximating the **expected** distance $E||r^\top x||_2$ will diminish (resulting in increased error between them). Therefore, the distance $||r^\top x||_2$ of the *actual* matrix may not be accurately reflected by the *expected* distance $E||r^\top x||_2$, making it challenging to effectively estimate the actual classification performance of the actual matrix using the *expected* distance $E||r^\top x||_2$.
>
>
>
> [r1] D. Achlioptas. Database-friendly random projections: Johnson–Lindenstrauss with binary coins. J. Comput. Syst. Sci., 66(4):671–687, 2003.
>
> [r2] P. Li, T. J. Hastie, and K. W. Church. Very sparse random projections. in Proceedings of the 12th ACM SIGKDD international conference on Knowledge discovery and data mining, 2006.
>
> In the experiments (Figures 4-7),  we examined the classification performance against varying sparsity $k$, both for $\ell_1$ norm and $\ell_2$ norm. It can be seen that both norms exhibit similar performance trends; and in other words, similarly to $\ell_1$ norm,  the classification on $\ell_2$ norm is also related to the sparsity $k$.  Nonetheless, as previously discussed, analyzing the classification performance with the $\ell_2$ norm through $E||r^\top x||_2$ (or $Var(||r^\top x||_2)$) presents challenges.
>
> In summary, instead of the $\ell_2$ norm, we have opted  the $\ell_1$ norm to examine the distance preservation property and classification performance of sparse matrices, mainly due to its analytical tractability. Empirically, as mentioned above, the $\ell_2$ norm yields similar results, while the theoretical analysis for it may be deferred to future research, given the existing challenges.
>
>
> **C2:**  I find the argument that larger distances improves classification accuracy to be too-general and vague. As per the authors argument, $\ell_2$ distances are preserved by sparse matrix-base random projection. Does that mean the classification accuracy is the same before and after projections as far as $\ell_2$  is concerned? It would be useful if the authors could be more specific about how maximising the $\ell_1$ distance (not just any distance) leads to improved classification accuracy.
>
> **R2:**  Thank you. As early discussed, random projection can only maintain the expected $\ell_2$ distance $E||r^\top x||_2$, rather than the  distance $||r^\top x||_2$ of *actual* matrices. After random projection,  classification accuracy tends to decline  due to the distance errors.
>
> As explained in the previous response R2, the argument that larger  pairwise distances between projected data points tend to yield better classification performance, stems from Principle Component Analysis (PCA), which is a cornerstone principle of PCA. This property has been widely validated in various PCA applications. A typical example is face recognition [r3]. Studies have demonstrated that the projection onto larger principle components (resulting in larger distances) tends to produce more distinct features, and then yield better classification performance.  For more evidence, please see: \url{https://en.wikipedia.org/wiki/Eigenface}.
>
> [r3] M.A. Turk and A.P. Pentland. Face recognition using eigenfaces. In Proceedings. 1991 IEEE Computer Society Conference on Computer Vision and Pattern Recognition, pp. 586–591, 1991.

---

> > ### Comment · Reviewer_gLeX · 2025-01-10
> >
> > $\textbf{1. More clarification regarding R1}$
> >
> > If the variance increases with sparsity and the expected value $\mathbb{E}[\left |r^Tx \right |]$ may be far away from the actual distance between projected data at low sparsity, then it is not necessary that low sparsity actually leads to better classification performance due to high variance. How do we interpret the point 1 (P1) under remarks of Theorem 1?
> >
> > The conclusion is that the $\mathbb{E}[\left |r^Tx \right |]$ attains high values at low levels of sparsity and hence results in better classification accuracy. However, low levels of sparsity has high variance and hence it is challenging to effectively estimate the actual classification performance use the expected distance $\mathbb{E}[\left |r^Tx \right |]$. How can these seemingly contradictory statements be reconciled?
> >
> > $\textbf{2. More clarification regarding R2}$
> >
> > The principle behind PCA is that the eigen-values represent the variance of data in the respective eigen-directions, and hence projecting the data in the principal eigen-directions preserves information and more useful for downstream application. Is the notion of high variance along the principal  eigen-directions interchangeably used to indicate larger pair-wise distances?

---

> ### Author Response · Authors · 2025-01-09
> **Response to Reviewer gLeX**
>
> **C3:** The equation (3) is unclear. I agree to its validity when each element of $h$, $h'$ are samples from a univariate Gaussian distribution with mean $\mu$ and variance $\sigma^2$. Can the authors elaborate on why it still holds when each entry is itself from a Gaussian mixture model with say $M$ components?
>
> **R3:** Thank you.  In the previous Response R3, we have mentioned that given two original data points $h$ and $h'$, each corresponding pair of  elements  $h_i$ and $h'_i$ will be  drawn either from the same distribution, or not. To reflect this fact, we model  $h_i$ and $h'_i$  using a **two**-component Gaussian mixture:
>
> $\Sigma_{i=1}^{2} w_i N(\mu_i, \sigma^2)$,  $\mu_1\neq\mu_2$, $w_1+w_2=1$，$w_i\geq 0$.
>
> When  $h_i$ and $h'_i$ are both drawn from $N(u_1, \sigma^2)$ (or $N(u_2, \sigma^2) $), their difference $x_i=h-h' \sim N(0, 2\sigma^2)$; otherwise, $x_i\sim N(\mu=\mu_1-\mu_2,2\sigma^2)$ or $N(-\mu=\mu_2-\mu_1, 2\sigma^2)$.  This result in a **three**-component Gaussian mixture for $x_i$, as described in (3). We will include the above discussion in the revised manuscript.
>
> Moreover, by the above analysis, it can be seen that  if we increase the number  $M_h$ of mixture components of $h_i$ and $h'_i$,    the component number  $M_x$ of their difference $x_i$ may change, rather than fixing to three. As discussed above, in the paper we only need to consider the case of $M_h=2$ and $M_x=3$.

---

> ### Author Response · Authors · 2025-01-10
> **Response to Reviewer gLeX**
>
> Dear Reviewer gLeX,
>
> Thank you for your further comments. Below we reply to them one by one.
>
> **C1:**  The relation between ${E}|{r^\top x}|$ and ${Var}(|{r^\top x}|)$.
> 1)  If the variance increases with sparsity and the expected value ${E}|{r^\top x}|$ may be far away from the actual distance between projected data at low sparsity, then it is not necessary that low sparsity actually leads to better classification performance due to high variance. How do we interpret the point 1 (P1) under remarks of Theorem 1?
>
> 2) The conclusion is that the ${E}|{r^\top x}|$ attains high values at low levels of sparsity and hence results in better classification accuracy. However, low levels of sparsity has high variance and hence it is challenging to effectively estimate the actual classification performance use the expected distance  ${E}|{r^\top x}|$. How can these seemingly contradictory statements be reconciled?
>
>
>
> **R1:**
> 1) In the analysis of  the expected distance ${E}|{r^\top x}|$, the influence of the variance ${Var}(|{r^\top x}|)$  in (5) should be considered. Statistically, a lower  variance ${Var}(|{r^\top x}|)$ indicates a higher probability  that the actual distance $|{r^\top x}|$ of a single matrix closely approximates its expected value ${E}|{r^\top x}|$. Also, this implies a higher consistence between theoretical and practical results. By computing (5),  as detailed in **the supplement**,  we observe that the larger ${E}|{r^\top x}|$ tends to correspond to smaller $Var|{r^\top x}|$. This implies that in P1, the maximum  ${E}|{r^\top x}|$ achieved at $k=1$ is relatively close to the distance $|{r^\top x}|$ of actual matrices.
>
> 2)  As $k$ increases, by computing (5),  we observe a trend similar to ${E}|{r^\top x}|$: ${Var}(|{r^\top x}|)$  tends to quickly converge to a constant value.  This suggests that ${Var}(|{r^\top x}|)$ varies minimally across different $k$ values. Therefore,  the probability of $|{r^\top x}|$ approximating ${E}|{r^\top x}|$ remains consistent for various $k$, enabling us to use ${E}|{r^\top x}|$ to reasonably estimate and compare the distances $|{r^\top x}|$ of actual matrices across different $k$.
>
> **C2:**  The principle behind PCA is that the eigen-values represent the variance of data in the respective eigen-directions, and hence projecting the data in the principal eigen-directions preserves information and more useful for downstream application. Is the notion of high variance along the principal eigen-directions interchangeably used to indicate larger pair-wise distances?
>
> **R2:** Yes. Larger pairwise distances generally result in higher variances.  Note that in Definition 1, we require that sparse matrices $R$ with varying sparsity $k$ maintain the $\ell_2$ distance preservation property, namely $E||Rx||_2=E||x||_2$. This ensures that the projections of these matrices remain consistent in magnitude across different sparsity $k$.

---

> ### Comment · Reviewer_gLeX · 2025-01-11
>
> Thank you for your responses. It certainly helps in understanding the contributions better.
>
> It is seen that the accuracy for both $\ell_1$ and $\ell_2$ distances are similar. The Johnson–Lindenstrauss lemma states that $\ell_2$ distances are preserved using random projections even when $m = O(log(n))$. However, the current analysis requires that $m = O(\sqrt{n})$, which is much higher.
>
> **Question**: Why not project the data onto $O(log (n))$ dimension, use $\ell_2$ distance for classification, and achieve comparable accuracy, in lieu of sparsely projecting data onto a higher $O(\sqrt{n})$ dimension and classify them using $\ell_1$?

---

> > ### Author Response · Authors · 2025-01-11
> > **Response to Reviewer gLeX**
> >
> > Dear Reviewer gLeX,
> >
> > Thank you for your interest and helpful feedback on our work. It has really helped us improve the manuscript.
> >
> > **Comment:**  Why not project the data onto $O(log(n))$ dimension,  use $\ell_2$ distance for classification, and achieve comparable accuracy, in lieu of sparsely projecting data onto a higher  dimension $O(\sqrt{n})$  and classify them using  $\ell_1$?
> >
> > **Response:**  Our results differ in terms of the projection dimension condition because of the differences in the matrices we studied. In the $\ell_2$ analysis  [r1], the author derived $m\geq O(log(n))$  for sparse \{0,$\pm 1$\} matrices where the proportion of $\pm 1$ is **1/3**. In contrast, in our $\ell_1$ analysis, we considered matrices where the number of $\pm 1$ entries per row can vary widely, from 1 (extremely sparse) to $n$ (dense). Additionally, it is worth noting that pursuing excessively low compression ratios $m/n$ may *not* be practical, as it can result in significant feature loss and subsequently degrade the performance of subsequent tasks. For example, in random projection-based face recognition [r2], as shown in Figures 8 and 9, the face recognition rate drops sharply from 90% to 50% when the feature dimension $m$ is reduced from 504 to 30.
> >
> > [r1] D. Achlioptas. Database-friendly random projections: Johnson–Lindenstrauss with binary coins. J. Comput. Syst. Sci., 66(4):671–687, 2003.
> >
> > [r2] Wright J, Yang A Y, Ganesh A, et al. Robust face recognition via sparse representation[J]. IEEE transactions on pattern analysis and machine intelligence, 2008, 31(2): 210-227.
> >
> > In the manuscript, we have opted to investigate  the $\ell_1$ norm for two major reasons. 1) Similar to the $\ell_2$ norm, the $\ell_1$ norm holds significant importance in machine learning [r3]. It is valuable to examine the influence of sparsity (in sparse matrices) on the expected $\ell_1$ distance $E|r^\top x|$ both in theory and practice.   2) By examining the variation of the expected $\ell_1$ distance $E|r^\top x|$ across different matrix sparsity $k$, as shown  in P1 and P2, we can reasonably predict the fluctuation in classification performance. In contrast, as discussed in our introduction, achieving this through an analysis of the $\ell_2$ distance seems challenging.
> >
> > [r3] Bo Brinkman and Moses Charikar. On the impossibility of dimension reduction in $\ell_1$. Journal of the ACM, pp. 766–788, 2003.

---

> > > ### Comment · Reviewer_gLeX · 2025-01-13
> > >
> > > Projection of data to lower dimensional is different from any subsequent tasks, say classification. The classification performance do primarily depend on the type of classifier used on the projected data. For instance, one can project the data using (i) Gaussian matrices, (ii) matrices where the sparsity proportion is $\frac{1}{3}$ or (iii) the number of $\pm$ entries per row can vary widely, from 1 (extremely sparse) to $n$ (dense). Once the data is projected, any type of classification methods such as $\ell_1$, $\ell_2$, SVM's, deep learning, etc. can be subsequently used.
> > >
> > > I do find the analysis with respect to how sparsity impacts $\mathbb{E} [\left|r^T x \right|]$ and $Var [\left|r^T x \right|]$ to be quite interesting, and there are definite merits to it. But the connection to classification appears more of an observation on the basis that larger pairwise distances generally yields higher classification accuracy. To this end, I second the remark made by the action editor that "there are no theoretical results explaining why classification accuracy should improve by maximising the $\ell_1$ distances".

---

> > > > ### Author Response · Authors · 2025-01-13
> > > > **Response to Reviewer gLeX**
> > > >
> > > > Thank you for approving our contribution. As discussed earlier,  the principle that larger pairwise distances (variances) between projected points tends to lead to better classification, stems from  PCA, serving as  a cornerstone of PCA. This principle has gained widespread recognition and validation through PCA's numerous applications, despite as  the author and editor noted, it appears to lack rigorous theoretical proof [r1]. **Given its broad acceptance in PCA research, we believe the adoption of this principle in our research is equally reasonable.**
> > > >
> > > > [r1] Ian T Jolliffe and Jorge Cadima. Principal component analysis: a review and recent developments. Philosophical transactions of the royal society A: Mathematical, Physical and Engineering Sciences, 374(2065): 20150202, 2016.

---

### Review · Reviewer_dYV1 · 2025-01-06

**Summary Of Contributions:**

The paper explores the effectiveness of sparse matrix based random projections for classification problems. In particular, it aims to analyze the trade-off between the number of non-zero ($\{-1,1\}$) entries in the projection matrix and its impact in the classification performance. In particular, it studies the $\ell_1$ distance between the projected data points in two distribution settings: Gaussian and two-point distributions. In such settings, the paper theoretically and numerically shows that with only few non-zero entries per row, good classification performance can be achieved when compared with other more dense matrices.

**Audience:**

Yes

**Claims And Evidence:**

No

**Requested Changes:**

1. In Theorem 3.1, the paper has an analysis for $k\rightarrow\infty$ setting. I am wondering what does this setting implies because $k$ has an upper bound of $n$. So does $k$ and $n$ both tends to infinity? If yes, at what rates?

2. I also did not get the significance of point P2 in Theorem 1 and its remark. The utility of analysis in (8) and (14) is also unclear.

3. Proper justification should be provided for this statement: "Then we can say that for two discriminative data distributions, the best classification performance should be able to be achieved using very sparse random matrices with sparsity"

4. In (P3) (page 6), it is unclear what is meant by the statement:  "Compared to the theoretical result P1, the numerical result relaxes the upper bound of $p$ from .... "

5. In P4, what does "convergence speed is fast" imply?

6. Similarly, the statement "Var[|r\top x|] exhibits a changing trend opposite to that of E[r^\top x]" should be explained. In addition, why is "smaller variances are favorable to classification"?

7. How was the Gaussian matrix (used for random projection baseline in Section 5) generated?

8. Why were the original dimensions of YaleB and Newsgroups datasets reduced? They do not seem too big (and smaller than the dimensions in ALL dataset)?

**Strengths And Weaknesses:**

Strengths:
The paper focuses on an interesting problem related to random sparse projections. The paper has analyzed the $\ell_1$ distance between the projected data points in two distribution settings: Gaussian and two-point distributions.

Weakness:
1. Several statements are unclear. Please see the requested changes section.
2 The paper should be proof-read and grammatical mistakes should be resolved. While a major revision was suggested previously, it does not seem that sufficient effort was put in this direction.

---

> ### Author Response · Authors · 2025-01-07
> **Response to Reviewer dYV1**
>
> Dear Reviewer  dYV1,
>
> Thank you very much for dedicating your valuable time to review our manuscript. Below we reply to the comments one by one.
>
> **C1:** In Theorem 3.1, the paper has an analysis for $k\to \infty$ setting. I am wondering what does this setting implies because $k$ has an upper bound of $n$. So does $k$ and $n$ both tends to infinity? If yes, at what rates?
>
> **R1:** Thank you. Given the relationship $k\leq n$, we implicitly assume that $n\to\infty$ when considering $k\to\infty$. In our analysis, there are no additional constraints on the rates at which $k$ and $n$ approach infinity, aside from the inherent condition $k\leq n$. Our numerical results, presented in Figures 1 and 3, suggest that the limit in (6) can be approximated even when $k$ is relatively small, such as in the range of dozens, rather than requiring $k$ to be very large or tending to infinity.
>
> **C2:** I also did not get the significance of point P2 in Theorem 1 and its remark. The utility of analysis in (8) and (14) is also unclear.
>
> **R2:** Thank you.  Roughly speaking, in P2 we aim to prove that the expected $\ell_1$ distance $E\|r^\top x\|$ does not vary significantly across different values of $k$; This finding allows us to conclude that using smaller $k$ values (corresponding to sparse matrices) yields comparable performance to larger $k$ values (corresponding to dense matrices).
>
> For this purpose, we first establish that $E\|r^\top x\|$ tends to a constant $2\sqrt{q/\pi}$, when $k\to \infty$ in (6). Subsequently, in (7), we derive an upper bound for the converge error, which scales as $1/\sqrt{k}$. This implifies that that difference in $E\|r^\top x\|$ between any $k$ values within the interval $k\in[K,\infty]$, will be less than $O(1/\sqrt{K})$. Notably, this difference can achieve relatively small values, even when $K$ is not very large. Consequently, we conclude that using smaller $k$ values (sparse matrices) yields performance that is comparable to using larger $k$ values (dense matrices).
>
> **The utility of analysis in (8) and (14):** In practical matrix generation scenarios, we often need to determine an appropriate (smallest) matrix sparsity $K$ (as mentioned above in $k\in[K,\infty]$), based on a specified or desired convergence error ratio $\eta$. The value of $K$, which serves as the lower bound for $k$, can be derived from equations (8) and (14). This allows us to tailor the sparsity of the matrix to achieve the desired  convergence accuracy. Overall, (8) and (14) are crucial  results,  provided by the suggestion of the previous Reviewer **wNEK**.
>
> **C3:** Proper justification should be provided for this statement: "Then we can say that for two discriminative data distributions, the best classification performance should be able to be achieved using very sparse random matrices with sparsity"
>
> **R3:** Thank you. The theoretical result P1 has been validated in both the numerical results (Figure 1a and Figure 3a) and experimental results (Figure 4c and Figure 11c).
>
>
> **C4:** In (P3) (page 6), it is unclear what is meant by the statement: "Compared to the theoretical result P1, the numerical result relaxes the upper bound of p from .... "
>
> **R4:** Thank you. The numerical results indicate that to achieve the maximum value of $E\|r^\top x\|$ at $k=1$, the condition $p\in[0,1/3)$ is sufficient, which is broader than the theoretical requirement $p\in[0,0.188)$ derived from P1. Recall that a wider range of $p$ allows for a larger space of data as modeled by (3). This suggests that the desired property of maximizing $E\|r^\top x\|$ at $k=1$ can be achieved over a wider range of $p$ values than what was theoretically predicted. We will include this discussion in the revised manuscript.
>
>
> **C5:** In P4, what does "convergence speed is fast" imply?
>
> **R5:** Thank you. What we mean is that $E\|r^\top x\|/(\mu\sqrt{n/m})$ quickly converges to a constant value with the increasing of $k$.

---

> ### Author Response · Authors · 2025-01-07
> **Response to Reviewer dYV1**
>
> **C6:** Similarly, the statement "Var[|r\top x|] exhibits a changing trend opposite to that of E[r^\top x]" should be explained. In addition, why is "smaller variances are favorable to classification"?
>
> **R6:** Thank you.  What we intend to convey is that as the expected value $E\|r^\top x\|$ increases, the variance $Var（\|r^\top x\|)$ tends to decrease. For clarity, we have revised this section of the manuscript as follows.
>
> In the analysis of  the expected distance ${E}|{r^\top x}|$ analyzed above, the influence of the variance ${Var}(|{r^\top x}|)$  in (5) should be considered. Statistically, a lower  variance ${Var}(|{r^\top x}|)$ indicates a higher probability  that the actual distance $|{r^\top x}|$ of a single matrix closely approximates its expected value ${E}|{r^\top x}|$. Also, this implies a higher consistence between theoretical and practical results.    By computing (5),  we observe a trend similar to ${E}|{r^\top x}|$:  as $k$ increases, ${Var}(|{r^\top x}|)$  tends to quickly converge to a constant value (see the supplement for further details). This suggests that ${Var}(|{r^\top x}|)$ varies minimally across different $k$ values. Therefore,  the probability of $|{r^\top x}|$ approximating ${E}|{r^\top x}|$ remains consistent for various $k$,  enabling us to use ${E}|{r^\top x}|$  to reasonably  estimate and compare the  distances $|{r^\top x}|$ of actual matrices across different $k$
>
> **C7:** How was the Gaussian matrix (used for random projection baseline in Section 5) generated?
>
> **R7:** Thank you. We generated the Gaussian matrix by drawing its each element i.i.d. from $N(0,1)$. We will detail this in the revised manuscript.
>
>
> **C8:** Why were the original dimensions of YaleB and Newsgroups datasets reduced? They do not seem too big (and smaller than the dimensions in ALL dataset)?
>
> **R8:** Thank you. In the experiments, we have tested the KNN and SVM classification performance for the random matrices with varying sparsity $k\in[1,30]$, and projection ratio $m/n \in$\{1\%,10\%,50\%\} across four different datasets. To achieve reliable classification performance, we conducted hundreds of repeated experiments to obtain average results. This is a time-consuming process. To reduce simulation time, we decreased the data dimensions by *integer multiples*, reducing them to the thousands. It is worth noting that these dimension values were not intentionally chosen. Our results are valid for higher data dimensions, as demonstrated in our theoretical analysis.
>
> **C9:**  The paper should be proof-read and grammatical mistakes should be resolved.
>
> **R9:** Thank you. We will thoroughly proofread the manuscript and correct all grammatical mistakes before submitting the revised version.

---

> > ### Comment · Reviewer_dYV1 · 2025-02-10
> > **Response to authors' rebuttal**
> >
> > Dear authors,
> >
> > Thanks a lot for your reply. In the above response, major concerns have not been addressed. Please see the details below.
> >
> > - Regarding C2/R2: The claim that the analysis for the setting $k\rightarrow\infty$ shows that "expectation does not vary significantly across different values of k" is technically incorrect. This is because a given problem setting has $n$ as fixed and $k$ could be varied from $1$ to $n$. This $n$ could be small, in which case $k$ will never tend to $\infty$. In fact, in many experiments discussed in the paper, the value of $n$ is quite small.
> >
> > - Regarding C3/R3: Statement P1 does not theoretically justify the statement "Then we can say that for two discriminative data distributions, the best classification performance should be able to be achieved using very sparse random matrices with sparsity". It simply states that when $p\leq 0.188$, the expectation has maximum value with $k=1$. How does the expectation is theoretically related to the generalization error in classification error is unclear.
> >
> > - Regarding C4/R4: One cannot justify that the condition $p\in[0,1/3)$] is sufficient simply on the basis of a few numerical/synthetic experiments unless it can be shown that the experiments model all possible real-world settings.
> >
> > - Regarding C5/R5: while "fast" and "quickly" have same semantics w.r.t. convergence, I was more interested if the degree of quickness can be quantified.
> >
> > - Regarding C6/R6: Is this statement "a lower variance $Var(|r^\top x|)$ indicates a higher probability that the actual distance $|r^\top x|$ of a single matrix closely approximates its expected value $E[|r^\top x|]$." always true?
> >
> > - Regarding C7/R7: The parameters of the Gaussian distribution, which parameterized the random projection baseline, was not tuned. Hence, it seems that the baseline is at a disadvantage.
> >
> > - Regarding C8/R8: It is unclear from the draft the number of runs over which average results were obtained. How much time does a single run of these experiments take?

---

> ### Author Response · Authors · 2025-02-10
> **Response to Reviewer dYV1**
>
> Dear reviewer dYV1,
>
>
> Thank you for your further feedback. First of all, **we would like to emphasize that our  main theoretical results presented in Theorems 1 and 2 are *correct*, and the conclusions derived from these theoretical results are *reasonable*.** Let us address each concern as follows:
>
> **Response to "Regarding C2/R2":** In P1 (Theorem 1), we establish the convergence relation Eq. (6) under the condition of $k\rightarrow \infty$ ($n\rightarrow \infty$)， and demonstrate that the convergence error decreases with a speed of $O(1/\sqrt{k})$  in Eq. (7). This implies that we can achieve a  small convergence error when $k$ is relatively small, rather than approaching infinity. The small convergence error achieved with relatively small $k$ allows us to reach the conclusion that “expectation does not vary significantly across different values of $k$”, even when $k$ (and $n$) is small as evidenced in our simulations and experiments. Moreover, it is worth noting that the *actual* decreasing speed of convergence errors should be faster than $O(1/\sqrt{k})$, as the bound we derive in Eq. (7) contains slacks.
>
>
> **Response to "Regarding C3/R3":**    As mentioned in our Introduction, our argument that "the larger pairwise distance between projected data points (namely, the larger expected value $E|r^\top x|$) is expected to lead to the better classification performance", arises  from the research of principal component analysis (PCA) [(Jolliffe, 2002)]. In PCA, it is argued that projecting data onto  *larger* principal components results in *larger* variances (achievable by larger pairwise distances) among the projected data points; and these *larger* variances are beneficial as they tend to capture more of the variation (i.e., statistical information) inherent in the original data [(Jolliffe & Cadima, 2016)], thereby achieving better classification performance. This principle has been widely recognized and validated in various PCA applications, such as face recognition [ (Turk & Pentland, 1991)].
>
>
>
> **Response to “Regarding C4/R4”:** Notice that the condition $p<0.188$ derived in P1 (Theorem 1)  is **sufficient**  but not **necessary**. To  investigate the *actual* bound, we conduct numerical simulations in P3  and find that the bound can be extended from $p<0.188$ to $p\in[0, 1/3)$. The value of our simulation result   $p\in[0, 1/3)$  lies in providing more guidance for practical applications (despite its inaccurateness), while it  does *not* affect the correctness of our theoretical result $p<0.188$.
>
> **Response to “Regarding C5/R5”:**  The decreasing speeds (i.e. $O(1/\sqrt{k})$) of the convergence errors in Eqs.(6) and (12) have been given in Eqs (7) and (13), respectively.
>
> **Response to “Regarding C6/R6”:**  By Chebyshev's inequality,  it is known that：  with *higher* probability, a random variable (i.e. $|r^\top x|$) will take values close to its expectation (i.e. $E|r^\top x|$), as its variance (i.e. $Var(|r^\top x|)$) *decreases*.
>
> **Response to “Regarding C7/R7”:**  Notice that the Gaussian matrix with elements i.i.d. drawn from $N(0,1)$ has been widely used in random projection as a state-of-the-art method, since it holds the distance preservation property. Technically, there is no need to adjust its mean or variance; and no existing work has done so.
>
>
>
> **Response to “Regarding C8/R8”:** For reproducibility, we have provided the code at the link: https://anonymous.4open.science/r/The-Sparse-Matrix-Based-Random-Projection-An-Analysis-of-Matrix-Sparsity-for-Classification-7D00
>
> For the YaleB dataset, SVM classification takes about 1000 minutes to produce results across different matrix sparsity $k\in[1,30]$ and compression ratios $m/n \in$ \{1\%, 10\%, 50\%\}, running on an Intel i9-10980XE CPU platform. The simulation time for other datasets will vary slightly  due to differences in dataset size and data dimensionality.
>
> As described in the paper, we conduct binary classification by evaluating all possible class pairs within each dataset. For example, in the YaleB dataset, which includes 38 individuals, we perform binary classification on 703 unique pairs of individuals. For each pair, we randomly split the data into training and testing sets 20 times. For each split, we repeat the random projection and classification process 5 times. The final binary classification accuracy is obtained by averaging the results from these repetitions.

---

> ### Comment · Reviewer_dYV1 · 2025-02-10
> **Response to "Response to "Regarding C2/R2""**
>
> Let us consider the bound (8) derived from (7). In order for the above mentioned claim to be true, one should be able to obtain a low convergence ratio error ($\eta$) in (8) with reasonably low value of $k$. As discussed in P4, let us consider p = 2/3 (as considered in Fig 1(b) and Fig 1(d)). This implies q = 1/6. If one consider $\eta = 0.1$, (8) implies that k>= 1523. With p = 1/3 (as considered in Fig 1(a) and Fig 1(c)), we require k >= 761 in order to obtain $\eta = 0.1$. Both are significantly high values of $k$. If one requires a lower $\eta$ than 0.1, the lower bound on $k$ would be even higher (due to inverse quadratic dependence). Thus, the claim in the above response "The small convergence error achieved with relatively small k allows us to reach the conclusion that “expectation does not vary significantly across different values of k"" is incorrect.

---

> > ### Comment · Reviewer_dYV1 · 2025-02-10
> > **Response to "Response to “Regarding C4/R4”"**
> >
> > Please note that my query was not on the theoretical correctness of the result related to the range $p<0.188$. My query was on how it was proved that the range $p\in[0,1/3)$ was considered sufficient. This is because the claim given in R4 above ("the condition $p\in[0,1/3)$ is sufficient") should be formally proved.

---

> > > ### Comment · Reviewer_dYV1 · 2025-02-10
> > > **Response to "Response to "Regarding C3/R3""**
> > >
> > > Based on the authors' response, it seems that conclusions/observations from previous works were taken as a motivation for the study, instead of providing a concrete theoretical relationship between the expectation and the generalization error in classification setting. However, in case my understanding is incorrect, I would request the authors to provide the details in the draft for completeness (perhaps in supplementary).
> > >
> > > Empirically, it appears that in a few experimental results (e.g., Fig 5 (b)), sparse random matrices (with only one or a few dozen nonzero entries per row) does not achieve comparable classification results compared to denser matrices. In addition, while the analysis in Sections 3 and 4 has highlighted that $k=1$ attains maximum pairwise distance between projected data points. However, $k=1$ performs poorly on Newsgroup and MNIST datasets (2 out of 4 datasets). Hence, it is unclear whether the above discussed motivation necessarily translates into lower generalization performance.

---

> > > > ### Comment · Reviewer_dYV1 · 2025-02-10
> > > > **Other responses**
> > > >
> > > > - Regarding discussion on C5/R5: I thank the authors for the clarification.
> > > >
> > > > - Regarding discussion on C6/R6: Accordingly to Chebyshev's inequality, $P(|X - E[X]| \geq k ) \leq Var(X)/k^2$. Will the claim hold if $k < \sqrt{Var(X)}$?
> > > >
> > > > - Regarding discussion on C7/R7: It would be great if appropriate references could be provided (and mentioned in the draft for the benefit of the reader)
> > > >
> > > > - Regarding discussion on C8/R8: Since training SVM was taking considerable amount of time, other classifiers where training time is lesser could have been tried.

---

> > > > ### Author Response · Authors · 2025-02-11
> > > > **Response to Reviewer dYV1  "Regarding C3/R3"**
> > > >
> > > > In the Introduction, we have outlined the major contributions of the work: 1) For the first time, we provide a theoretical analysis of the $\ell_1$ distance variation in sparse matrix-based random projections, specifically exploring how this variation changes with matrix sparsity. 2) Then by PCA , we analyze the impact of $\ell_1$ distance variation on the classification performance of the projected data.
> > > >
> > > > Regarding the performance of $k=1$:  Notice that according to Theorem 1, $k=1$ will achieve the maximum pairwise distance only when $p<0.188$ (P1).  In other cases (P2),  the pairwise distances of different values of $k$ tend to become comparable, as $k$ increase.
> > > >
> > > > As demonstrated in our experiments, the results align well with the theoretical analysis.

---

> ### Author Response · Authors · 2025-02-11
> **Response to Reviewer dYV1 "Regarding Other responses"**
>
> **Comment:** Regarding discussion on C6/R6: Accordingly to Chebyshev's inequality, $P(|X - E[X]| \geq k ) \leq Var(X)/k^2$. Will the claim hold if $k < \sqrt{Var(X)}$?
>
> **Response:**  Chebyshev's inequality holds for arbitrary $k>0$. Please see: https://en.wikipedia.org/wiki/Chebyshev%27s_inequality
>
>
> **Comment:** Regarding discussion on C7/R7: It would be great if appropriate references could be provided (and mentioned in the draft for the benefit of the reader)
>
>
> **Response:**  Notice that our work's focus is on sparse matrices rather than Gaussian matrices. In fact, Gaussian matrices have been extensively studied in early research [Dasgupta & Gupta, 1999; Achlioptas, 2003;Brinkman & Charikar, 2003]. These works have been mentioned in our Introduction.
>
>
> **Comment:** Regarding discussion on C8/R8: Since training SVM was taking considerable amount of time, other classifiers where training time is lesser could have been tried.
>
>
> **Response:**  Note that we have provided the **total time** (1000 minutes) required for SVM to generate **all** experimental results for the YaleB dataset across various sparsity levels ($k$) and compression ratios ($m/n$). However, if we perform **a single** binary classification task involving two classes of samples, SVM takes only about **0.005 seconds**.
>
> For the sake of generality, we have examined two fundamental classifiers in the paper: the nearest neighbor classifier and the support vector machine (SVM). Both are simple yet representative.
>
> For reproducibility, we have provided the code at the link: https://anonymous.4open.science/r/The-Sparse-Matrix-Based-Random-Projection-An-Analysis-of-Matrix-Sparsity-for-Classification-7D00

---

> ### Author Response · Authors · 2025-02-11
> **Response to Reviewer dYV1 "Regarding C4/R4"**
>
> We agree that it is challenging to reach **an exactly correct** conclusion about $p\in(0,1/3]$  solely through numerical simulations, as it is impossible to enumerate all possible real-world data scenarios.
>
> However, as previously discussed, our numerical simulation result  $p\in(0,1/3]$ serves as a complement to the theoretical finding  $p<0.188$ (derived by a sufficient but not necessary condition). **This combination aims to provide practical guidance for real-world applications, bridging the gap between theory and practice.**

---

> > ### Comment · Reviewer_dYV1 · 2025-02-11
> > **Response to authors "Regarding C4/R4"**
> >
> > While guidance via numerical simulations always adds value, in the current context, the relevant simulations (Figure 1(a) and Figure 1(c)) do not capture much variability in input data. Overall, the number of simulations seems quite low to draw empirical conclusion with reasonable confidence.

---

> > > ### Author Response · Authors · 2025-02-11
> > > **Response to Reviewer dYV1 "Regarding C4/R4"**
> > >
> > > We agree with the reviewer that drawing a rigorous conclusion solely based on a finite number of numerical simulations and experiments on real data is challenging.
> > >
> > > However, we kindly request the reviewer to note that **our major contribution lies in the theoretical results presented in Theorems 1 and 2**. The numerical and experimental results are provided to further validate and complement these theoretical predictions.
> > >
> > > Interestingly, it has been observed that the numerical and experimental results outperform the theoretical expectations. Specifically, they achieve the desired performance convergence with smaller sparsity levels  $k$. **This is encouraging news for our research, as it suggests practical advantages beyond the theoretical guarantees**.

---

> ### Author Response · Authors · 2025-02-11
> **Response to   Reviewer dYV1 "Regarding C2/R2"**
>
> Regarding the lower bounds of $k$ derived by (7,8) and (13, 14) for respectively holding (6) and (12): as previously discussed, it is important to note that these bounds we derive theoretically are based on **sufficient rather than necessary** conditions. This means that  the bounds of $k$ we theoretically establish contain **slack**,  which  is common in theoretical analysis due to the inherent challenges of such analysis.
>
> For this reason, we further perform numerical simulations in Figures 1-3 to investigate the *actual* bounds on $k$. It is found that small convergence errors can  be achieved by much smaller sparsity  $k$ ( at the level of *tens* as illustrated in Figures 1-3),  than theoretically expected (at the level of *thousands* as computed by the reviewer).  So, there is no contradiction between our simulation results and theoretical results.
>
> Combining the theoretical, simulation and experimental results,  it is *reasonable* to conclude that  "The small convergence error
>  can be achieved with relatively small $k$ and the expectation does not vary significantly with the increasing of $k$".

---

> ### Comment · Reviewer_dYV1 · 2025-02-11
> **Response to authors "Regarding C2/R2"**
>
> I agree that the bounds derived are sufficient rather than necessary. And because of this gap in the analysis, one may need very high $k$ value (rather than a small one as claimed in P4) to formally guarantee a low convergence ratio error (as detailed in my previous response).
>
> Empirically, the paper has some numerical simulations to show that in the experiments performed, a smaller $k$ may suffice. However, as discussed in C4/R4 discussion, the number of simulations seems quite low to draw empirical conclusion with reasonable confidence. The draft neither acknowledges the gap between the theoretical bound and the bound obtained via (a few) experiments nor discusses the utility of the high theoretical bound if the empirical bound is so low.

---

> > ### Author Response · Authors · 2025-02-11
> > **Response to Reviewer dYV1   "Regarding C2/R2"**
> >
> > **Comment:**  I agree that the bounds derived are sufficient rather than necessary. And because of this gap in the analysis, one may need very high  $k$ value (rather than a small one as claimed in P4) to formally guarantee a low convergence ratio error (as detailed in my previous response).
> >
> > Empirically, the paper has some numerical simulations to show that in the experiments performed, a smaller
> > $k$ may suffice. However, as discussed in C4/R4 discussion, the number of simulations seems quite low to draw empirical conclusion with reasonable confidence. The draft neither acknowledges the gap between the theoretical bound and the bound obtained via (a few) experiments nor discusses the utility of the high theoretical bound if the empirical bound is so low.
> >
> > **Response:**  Thank you for the comments. In the revised manuscript, we will emphasize that there is a gap between our theoretical bound and the underlying true bound observed in experiments. This clarification will help readers better understand the relationship between our theoretical predictions and empirical results.

---

> ### Comment · Reviewer_dYV1 · 2025-02-11
> **Response to authors "Regarding C3/R3"**
>
> The authors have not explained specific results such as poor performance in Fig 5 (b).
>
> Regarding the statement "Notice that according to Theorem 1, $k=1$ will achieve the maximum pairwise distance only when $p<0.188$  (P1)." : Is the empirical guidance of $p\in[0,1/3)$ be not applicable here?

---

> > ### Author Response · Authors · 2025-02-11
> > **Response to Reviewer dYV1 "Regarding C3/R3"**
> >
> > **Comment:** The authors have not explained specific results such as poor performance in Fig 5 (b). Regarding the statement "Notice that according to Theorem 1, will achieve the maximum pairwise distance only when  $p<0.188$ (P1)." : Is the empirical guidance of $p\in[0, 1/3)$  be not applicable here?
> >
> > **Response:**   Regarding the performance of Fig 5 (b), please see the first paragraph in the page 12 of the paper.

---

> ### Comment · Reviewer_dYV1 · 2025-02-11
> **Response to authors "Regarding Other responses"**
>
> Regarding Chebyshev's inequality: While Chebyshev inequality holds for arbitrary $k > 0$, if $k < \sqrt{Var(X)}$, then the Chebyshev's inequality gives the bound $P(|X - E[X] \geq k|) \leq 1$, which is not a very useful inequality in the context of the claim made by authors in an earlier response.
>
> Regarding references: the provided references have theoretical contributions in the area. It would be great if the authors could provide some recent works where the Gaussian matrix with elements i.i.d. drawn from $N(0,1)$ has been used in random projection as a state-of-the-art method.
>
> Regarding C8/R8: I thank the authors for the details. Since the SVMs take only about 0.005 seconds on average and NNC are usually quite efficient, it is unclear why YaleB and Newsgroups datasets' dimensions were reduced for computational reasons. The final dimensions of these datasets also seem missing.

---

> > ### Author Response · Authors · 2025-02-11
> > **Response to Reviewer dYV1 "Regarding Other responses"**
> >
> > **Comment:**  Regarding Chebyshev's inequality: While Chebyshev inequality holds $P(|X - E[X]| \geq \epsilon ) \leq Var(X)/\epsilon^2$ for arbitrary $\epsilon>0$,  if $\epsilon < \sqrt{Var(X)}$,  then the Chebyshev's inequality gives the bound  $P(|X - E[X]| \geq \epsilon ) \leq 1$, which is not a very useful inequality in the context of the claim made by authors in an earlier response.
> >
> > **Response:** We agree with the reviewer's interpretation of Chebyshev's inequality. Recall that our initial aim is to substantiate the assertion that "The smaller the variance $Var(X)$, the closer the value of $X$ to its expectation $E(X)$." From a statistical perspective, we generally establish this as follows. Let $\epsilon = 4\sqrt{Var(X)}$. By Chebyshev's inequality,  it can be deduced that $P(E(X)-4\sqrt{Var(X)}\leq X\leq E(X)+4\sqrt{Var(X)} )\geq 93.75$\%. This inequality suggests that as the variance $Var(X)$ decreases, $X$ will reside within a *narrower* interval centered at $E(X)$, indicating that $X$ is closer to $E(X)$.
> >
> > **Comment:** Regarding references: the provided references have theoretical contributions in the area. It would be great if the authors could provide some recent works where the Gaussian matrix with elements i.i.d. drawn from has been used in random projection as a state-of-the-art method.
> >
> > **Response:** In the area of random projection, as discussed in our Introduction, the commonly-used matrices mainly include Gaussian matrices with $N(0,1)$ elements and sparse $(0, 1)$ or $(0, \pm1)$ matrices.  **To the best of our knowledge, there is no theoretical or empirical research focusing on other distributed Gaussian matrices.**  **In the area of random projection, it is well-known that   $N(0,1)$-Gaussian matrices can well satisfy the distance preservation property, thus suitable for random projection**. For details, please see the references  we have provided in the Introduction.
> >
> >
> > **Comment：** Regarding C8/R8: I thank the authors for the details. Since the SVMs take only about 0.005 seconds on average and NNC are usually quite efficient, it is unclear why YaleB and Newsgroups datasets' dimensions were reduced for computational reasons. The final dimensions of these datasets also seem missing.
> >
> > **Response:**  This question has been answered in the previous response.  In the experiments, for each dataset, we have provided **a lot of** classification experiments (SVM and KNN)  by varying the matrix sparsity $k\in[1, 30] $ and the compression ratio $m/n\in$ \{1\%, 10\%, 50\%}. To obtain these results, SVM  takes about 1000 minutes for YaleB dataset, even though the data dimension is not large, merely 1200. If the reviewer is interested with the simulation time and experimental details, please examine and run the code available at the link:  https://anonymous.4open.science/r/The-Sparse-Matrix-Based-Random-Projection-An-Analysis-of-Matrix-Sparsity-for-Classification-7D00
> >
> > The data dimensions we have adopted for classification are 1200 , 3060,  7129, and 784,  respectively for YaleB, Newsgroups, AMLALL and MNIST. We will detail this in the revised manuscript.

---

### Comment · Reviewer_gLeX · 2025-01-03
**Concerns addressed in the revision**

By formally proving the result in the current submission, I see that the authors have addressed the comments raised by the reviewers of the earlier submission. I am willing to accept the paper. The paper is quite well written and easy to read.

Question to author (perhaps addressed earlier):
Any specific reason why the $\ell_1$ error was analysed and not $\ell_2$? Is it because after the $\sqrt \frac{n}{mk}$ scaling, the $\ell_2$ distances are preserved? If so, why?

---

### Comment · Action_Editor_qcf4 · 2025-01-07
**To authors**

Dear Authors,

We have received all the reviews for this submission, which was itself a revision of an earlier submission.

I have also gone through the reviews and the paper.

After reviewing the paper, I find that many critical concerns still remain in this submission, including:

1. There are no theoretical results explaining why classification accuracy should improve by maximizing the distances. Therefore, the empirical results do not make much sense apart from being mere observations.

2. There are doubts about the correctness of the theorems.

It is absolutely vital that your response and subsequent revisions sufficiently address the concerns of the reviewers and the above issues.

Best regards,

---

> ### Author Response · Authors · 2025-01-10
> **Response to Action Editor qcf4**
>
> Dear Action Editor qcf4,
>
> Thank you very much for dedicating your valuable time to review our manuscript. We have addressed all of the concerns raised by the reviewers, and submitted a revised manuscript. Below, we provide clarification on the two major points you have highlighted.
>
> **C1:** There are no theoretical results explaining why classification accuracy should improve by maximizing the distances. Therefore, the empirical results do not make much sense apart from being mere observations.
>
> **R1:** Thank you for raising this point, which is similar to the concern expressed by Reviewer gLeX. We reiterate our response as follows.  The argument that larger distances between projected data points tend to yield better classification performance, stems from principal component analysis (PCA) [r1]. In PCA, it is argued that projecting data onto  *larger* principal components results in *larger* variances (achievable by larger pairwise distances) among the projected data points; and these *larger* variances are beneficial as they tend to capture more of the variation (i.e., statistical information) inherent in the original data [r2], thereby achieving better classification performance. This principle has been widely recognized and validated in various PCA applications, such as face recognition [r3].
>
> [r1] Ian T Jolliffe. Principal component analysis. Springer, 2002.
>
> [r2] Ian T Jolliffe and Jorge Cadima. Principal component analysis: a review and recent developments. Philosophical transactions of the royal society A: Mathematical, Physical and Engineering Sciences, 374(2065): 20150202, 2016.
>
> [r3] M.A. Turk and A.P. Pentland. Face recognition using eigenfaces. In Proceedings. 1991 IEEE Computer Society Conference on Computer Vision and Pattern Recognition, pp. 586–591, 1991
>
> **C2:** There are doubts about the correctness of the theorems.
>
> **R2:** Thank you. We suspect you may be referring to the comments made by Reviewer dYV1, which primarily seek **clarification** regarding our theoretical and empirical results, rather than questioning their correctness. It is worth noting that our theoretical results have been thoroughly validated through numerical simulations illustrated in Figures 1, 2 and 3.

---

### Author Response · Authors · 2025-02-15
**General response to the editors and reviewers**

Dear Editors and Reviewers,

We sincerely express our gratitude to you  for dedicating your invaluable time and providing constructive feedback on our manuscript.

We have  revised the manuscript based on the comments received from all reviewers. The major changes have been highlighted in red for easy identification. To ensure reproducibility, we have included the code at the following link: https://anonymous.4open.science/r/The-Sparse-Matrix-Based-Random-Projection-An-Analysis-of-Matrix-Sparsity-for-Classification-7D00

Thank you once again for your contributions. We eagerly await the outcome of the review process.

Best regards,

The Authors

---

### Decision · Action_Editor_qcf4 · 2025-02-20

**Recommendation:** Reject

**Comment:**

I have thoroughly read and reviewed both the current and previous versions of the paper, including the reviews and the authors' responses. Based on this comprehensive evaluation (across total of 6 reviewers including both the submissions), I believe the paper does not meet the standards of TMLR for the following reasons:

1. The initial version contained many vague statements which prompted significant revisions. Unfortunately, the current version has not adequately addressed these issues. For instance, aside from Theorems 1 and 2 (which were present in the previous submission), the properties presented are mostly manipulations of those results without providing any substantial insights. Some questions are not clear, e.g., what are the implications and why those should be interesting.

2. The connection to improved classification generalization appears forced. It is unclear why one should lead to the other beyond an intuitive level.

3. If we disregard the properties and experiments (as they do not substantiate the claims), then there is little left in the paper. The entire paper could be condensed into a few pages, underscoring that the results are insufficient for publication in TMLR.

Regretfully, I have no choice but to reject the paper.

PS. I have taken the authors' concerns into account while making this decision.

**Audience:**

Appropriate.

**Claims And Evidence:**

The paper examines the ell_1 distance between data points and their random projections. It specifically claims that matrix sparsity is beneficial for classification performance and provides (?) theoretical results. However, the evidence supporting these claims is problematic. While certain analyses, such as Theorems 1 and 2, are interesting and well-received by reviewers, the rest of the evidence consists of empirical simulations. The claims regarding the effectiveness for classification generalization are based on limited empirical comparisons, and therefore, are not properly validated.

---

> ### Author Response · Authors · 2025-02-20
> **Response to the final comments made by  Action Editor qcf4**
>
> Dear Action Editor qcf4,
>
> We are writing to clarify  the final comments you made regarding our manuscript. While we deeply appreciate the time and effort you dedicated to evaluating our work, we must point out that the 3 comments you made are **incorrect**, as they fail to capture the **basic connections** between our theoretical results and empirical findings. We notice that Editor qcf4 has largely repeated the comments made by the 2 Reviewers dYV1 and gLex (invited in the second round of reviews). From their review comments, it is evident that the two reviewers are  inexperienced, having posed nearly 40 questions that are mostly basic in nature, and many of their opinions are **incorrect**, as we have already pointed out in our responses.  In contrast, the comments from the 4 reviewers in the first round of reviewing were **positive**, and the concerns they raised have been addressed in our revision. Given the open nature of the reviewing process, we would like to provide further clarification regarding the comments made by Action Editor qcf4.
>
> **Response to Comment 1:** Regarding the relations between our theoretical results (Theorems 1 and 2) and other numerical and experimental results, we wish to reiterate our clear statement as outlined in the paper. The **numerical results** are specifically provided to directly validate Theorems 1 and 2, ensuring their accuracy and reliability. Furthermore, the **experimental results** serve as evidence that our theoretical findings can indeed be utilized o predict classification performance, thereby highlighting the practical implications of our work.
>
> **Response to Comment 2:**  As we have consistently highlighted in our responses to the reviewers, our extension from theoretical results (Theorems 1 and 2) to classification performance is grounded in the principle of Principal Component Analysis (**PCA**). Specifically,  larger variances (achievable by larger pairwise distances) between projected data points typically lead to better classification performance.
>
> **Response to Comment 3:** As highlighted throughout the paper, our numerical and experimental results  **align with** our theoretical predictions. The experimental results are **sufficient**, having been conducted on four different datasets across various matrix sparsity levels and projection ratios. Furthermore, it has been observed that the numerical and experimental results even outperform our theoretical expectations. This is encouraging news for our research, as it suggests practical advantages beyond the theoretical guarantees.